# Impact of Multiple Radar reflectivity data assimilation on the numerical simulation of a Flash Flood Event during the HyMeX campaign

Ida Maiello[1,2], Sabrina Gentile[3,1], Rossella Ferretti[2], Luca Baldini[4], Nicoletta Roberto[4], Errico Picciotti[5,2], Pier Paolo Alberoni[6], Frank Silvio Marzano[1,2]

[1]Department of Information Engineering, Electronics and Telecommunications - Sapienza University of Rome, Rome, Italy

[2]CETEMPS, Department of Physical and Chemical Sciences - University of L'Aquila, L'Aquila, Italy

[3]Institute of Methodologies for Environmental Analysis, CNR IMAA, Potenza, Italy

[4]Institute of Atmospheric Sciences and Climate, CNR ISAC, Roma, Italy

[5]Himet s.r.l, L'Aquila, Italy

[6]Arpae Emilia Romagna - Servizio Idro-Meteo-Clima, Bologna, Italy

*Correspondence to*: Ida Maiello (ida.maiello@aquila.infn.it)

**Abstract.** An analysis to evaluate the impact of multiple radar reflectivity data with a three dimensional variational (3D-Var) assimilation system on a heavy precipitation event is presented. The main goal is to build a regionally-tuned numerical prediction model and a decision-support system for environmental civil protection services and demonstrate it in the central Italian regions, distinguishing which type of observations, conventional and not (or a combination of them) is more effective in improving the accuracy of the forecasted rainfall. In that respect, during the first Special Observation Period (SOP1) of HyMeX (Hydrological cycle in the Mediterranean Experiment) campaign several Intensive Observing Periods (IOPs) were launched and nine of which occurred in Italy. Among them, IOP4 is chosen for this study because of its low predictability regarding the exact location and amount of precipitation. This event hit central Italy on 14 September 2012 producing heavy precipitation and causing several damages to buildings, infrastructures and roads. Reflectivity data taken from three C-band Doppler radars running operationally during the event are assimilated using 3D-Var technique to improve high resolution initial conditions. In order to evaluate the impact of the assimilation procedure at different horizontal resolutions and to assess the impact of assimilating reflectivity data from multiple radars, several experiments using Weather Research and Forecasting (WRF) model are performed. Finally, traditional verification scores as accuracy, equitable threat score, false alarm ratio and frequency bias, interpreted analyzing their uncertainty through bootstrap confidence intervals (CIs), are used to objectively compare the experiments, using rain gauge data as benchmark.

*Keywords: radar data assimilation, WRF, 3D-Var, MET, bootstrap confidence intervals, HyMeX*

## 1 Introduction

In the last few years, a large number of floods caused by different meteorological events occurred in Italy. These events mainly affected small areas (few hundreds of square kilometers) making their forecast very difficult. Indeed, one of the

most important factors in producing a flash flood was found to be the persistence of the meteorological system over the
same area in the presence of specific hydrological conditions (the size of the drainage basin, the topography of the
basin, the amount of urban use within the basin, and so on), allowing for accumulating large amount of rain (Doswell et
al., 1996). In complex orography areas, such the Italian regions, this is largely due to the barrier effect produced by the
mountains, such as the Apennines. Moreover, the Mediterranean basin is affected by a complex meteorology, due to the
peculiar distribution of land and water and to the Mediterranean Sea temperature, which is warmer than that of the
European northern seas (Baltic Sea and North Sea). These factors may produce severe meteorological events: for
example, if precipitation persists over urbanized watersheds with steep slopes, devastating floods can occur in a
relatively short time.
The scientific community widely recognizes the need of numerical weather prediction (NWP) models to be run at high
resolution for improving very short term quantitative precipitation forecasts (QPF) during severe weather events and
flash floods. The combination of NWP models and weather radar observations has shown improved skill with respect to
extrapolation-based techniques (Sun et al., 2014). Nevertheless, the accuracy of the mesoscale NWP models is
negatively affected by the "spin-up" effect (Daley 1991) and is mostly dependent on the errors in the initial and lateral
boundary conditions (IC and BC, respectively), along with deficiencies in the numerical models themselves, and at the
resolution of kilometers even more critical because of the lack of high resolution observations, beside for radar data.
Several studies in the meteorological field have demonstrated that the assimilation of appropriate data into the NWP
models, especially radar (Sugimoto et al., 2009) and satellite ones (Sokol, 2009), significantly reduces the "spin-up"
effect and improves the IC and BC of the mesoscale models. Classical observations such as TEMP (upper level
temperature, humidity, and winds observations) or SYNOP (surface synoptic observations) do not have enough density
to describe for example local convection, while radar measurements can provide a sufficient density of data. Maiello et
al. (2014) showed the positive effect of the assimilation of radar data into the precipitation forecast of a heavy rainfall
event occurred in central Italy. The authors showed the gain by using assimilating radar data with respect to the
conventional ones. Similar results are obtained for a case of severe convective storm in Croatia by Stanesic and
Brewster (2016).
Weather radar has a fundamental role in showing tridimensional structures of convective storms and the associated
mesoscale and microscale systems (Nakatani, 2015). As an example, Xiao and Sun (2007) showed that the assimilation
of radar observation at high resolution (2km) can improve convective systems prediction. Recent researches in
meteorology have established that the assimilation of real-time data, especially radar measurements (radial velocities
and/or reflectivities), into the mesoscale NWP models can improve predicted precipitations for the next few hours. (e.g.
Xiao et al., 2005; Sokol and Rezacova, 2006; Dixon et al., 2009; Salonen et al., 2010).
The aim of this study is to investigate the potential of improving NWP rainfall forecasts by assimilating multiple radar
reflectivity data in combination or not with conventional observations. This may have a direct benefit also for
hydrological applications, particularly for real time flash flood prediction and consequently for civil protection
purposes. Major obstacles, that makes the assimilation of radar reflectivities into NWP models a challenging problem
both mathematically and physically, lie in the non-linear relation between radar reflectivity and precipitation intensity
as well as in the rapid evolution of mesoscale systems. While radial velocities observation operator is linear and based
directly on prognostic model variables (i.e. wind), the simulation of radar reflectivity is more challenging than radial
velocity, because the observation operator of radar reflectivity is highly non-linear and has a non-Gaussian error
probability density function.

The novelty of the paper is in exploring the impact on the high-resolution forecast of the assimilation of multiple radar reflectivity data in a complex orography area, such as central Italian regions, to predict intense precipitation. This aim is reached by using the IOP4 of the SOP1 in the framework of the HyMeX campaign (Ducrocq et al. 2014, Ferretti et al. 2014, Davolio et al. 2015). The SOP1 was held from 5 September to 5 November 2012; the IOP4 was issued for the central Italy target area on 14 September 2012 and it was tagged both as a Heavy Precipitation Event (HPE) and a Flash Flood Event (FFE). The reflectivity measured by three C-band weather radars was ingested together with traditional meteorological observations (SYNOP and TEMP) using 3D-Var to improve WRF model performance. So far, several studies about reflectivity data assimilation in heavy rainfall cases have been performed (e.g. Ha et al. 2011, Das et al. 2015) also including multiple radars data and in complex orography (e.g. Lee et al. 2010, Liu et al. 2013). However, this is the first experiment conducted on the Italian territory taking advantage of the reflectivity data collected by all the radars that cover central Italy.

The manuscript is arranged as follows. Section 2 provides information on the flash flood event and WRF model configuration. Section 3 presents the observations to be assimilated, the WRF 3D-Var data assimilation system, and the evaluation method used. The results are showed and assessed in the fourth and fifth Section. Summary and conclusions are reflected in the last Section.

## 2    Study area and model set up

Flash floods are still one of the natural hazards producing human and economic losses (Llasat et al. 2013). Moreover, an increasing trend of the occurrence of severe events in the whole Mediterranean area has been found by several authors (Hertig et al. 2012, Martin et al. 2013, Diodato and Bellocchi, 2014). These open issues drove the HyMeX programme (http://www.hymex.org) aims at a better understanding of the water cycle in the Mediterranean with focus on extreme weather events. The observation strategy of HyMeX is organized in a long-term (4 years) Enhanced Observation Periods (EOP) and short-term (2 months) Special Observation Periods (SOP). During the SOP1, that was held from 5 September to 5 November 2012 with the major aim of investigating still-unclear mesoscale meteorological mechanisms over the Mediterranean area, three Italian hydro-meteorological sites were identified within the Western Mediterranean Target Area (TA): Liguria–Tuscany (LT), northeastern Italy (NEI) and central Italy (CI). Several Intensive Observing Periods (IOPs) were issued during the campaign to document Heavy Precipitation Events (HPE), Flash Floods Events (FFE) and Orographic Precipitation Events (ORP).

### 2.1 Case study

During the day of 14 September 2012 a deep upper level trough entered the Mediterranean basin and deepened over the Tyrrhenian Sea slowly moving south eastward. A cut-off low developed over central Italy (Figure 1a, c) advecting cold air along the central Adriatic coast producing instability over central and southern Italy, and enhanced the Bora flow over the northern Adriatic Sea. Convection with heavy precipitations occurred in the morning of September 14 mainly along the central eastern Italian coast (Marche and Abruzzo regions), associated with the cut-off low over the Tyrrhenian Sea, producing flood in the urban area of Pescara where rainfall reached 150 mm in a few hours causing several river overflows, a landslide and many damages in the area of the city hospital. Progressive motion south-eastward of the cut-off and its filling (Figure 1b, d) gradually moved phenomena over south of Italy, even if some instability still remained over medium Adriatic until the afternoon of Saturday September 15. At the same time, a ridge

developed high pressure on the west part of West Mediterranean domain; this ridge slowly drifts eastwards during the weekend.

Figure 2, produced using DEWETRA operational platform, shows the interpolated map of 24h accumulated rainfall recorded from rain gauges network from September 14 to September 15 (00:00-00:00UTC) with a maximum accumulated rainfall on the highest peak of Abruzzo region (Campo Imperatore) approximately reaching 300 mm in 24 hours. DEWETRA (Italian Civil Protection Department, CIMA Research Foundation, 2014) is an operational web platform used by the Italian Civil Protection Department (DPC) and implemented by CIMA Research Foundation (http://www.cimafoundation.org/en/). DEWETRA allows synthesis, integration and comparison of information necessary for instrumental monitoring, models forecasting and to build real-time risk scenarios and their possible evolution. Rain gauges time series of some selected stations in Marche and Abruzzo regions, where most significant amount of rainfall is accumulated are presented in Figure 3: Fermo and Pintura di Bolognola (Marche region) respectively with nearly 130 mm in 24h (Figure 3a) and 180 mm in 24h (Figure 3b); Campo Imperatore, Atri and Pescara Colli (Abruzzo region) with respectively nearly 300 mm (Figure 3c), 160 mm (Figure 3d) and 140 mm (Figure 3e) in 24h. It is clearly shown (Figure 3) that the accumulation started around 02:00UTC of 14 September: in Fermo, Atri and Pescara Colli most of rainfall was concentrated in the first half of the day, whereas in Pintura di Bolognola and Campo Imperatore, precipitation fell all day long. The large amount of hourly precipitation for Atri and Pescara Colli respectively at 06:00UTC and 05:00UTC (red ovals in Fig. 3d and 3e) reaching 45mm/h, indicating convective precipitation, whereas rainfall at Campo Imperatore rain gauge (Fig. 3c) was much weaker but lasting longer which allowed for reaching an accumulated amount of approximately 300 mm in 24h.

Figure 4 shows the Vertical Maximum Intensity (VMI) reflectivity product from the Italian radar network (Vulpiani et al., 2008a) superimposed onto the Meteosat Second Generation (MSG) 10.8 μm image (in normalized inverted greyscale). A zoom over the central Italy target area highlights a line of convective cells along the Apennines in central Italy due to the western flow approaching the orographic barrier. VMI values above 45 dBZ are associated with intense precipitation that occurred during convective events.

**2.2 WRF model set up**

The numerical weather prediction experiments are performed in this work using the non-hydrostatic Advanced Research WRF (ARW) modeling system V3.4.1. It is a primitive equations mesoscale meteorological model, with terrain-following vertical coordinates and options for different physical parameterizations. Skamarock et al. (2008) provides a detailed overview of the model.

In this study, a one-way nested configuration using the *ndown* program is used: a 12 km domain (*263×185*) that covers central Europe and west Mediterranean basin (referred as D01) is initialized using the European Centre for Medium-Range Weather Forecasts (ECMWF) analyses at 0.25 degrees of horizontal resolution; an innermost domain, that covers the whole Italy (referred as D02), with a grid space of 3 km (*445×449*) using as BC and IC the output of the previous forecast at 12 km. Both domains run with 37 unequally spaced vertical levels, from the surface up to 100 hPa (Figure 5).

Taking into account that the performance of a mesoscale model is highly related to the parameterization schemes, the main physics packages used in this study are set as for the operational configuration (Ferretti et al., 2014) used at the

centre of Excellence CETEMPS. They include (Skamarock et al., 2008): the "New" Thompson et al. 2004 microphysics scheme, the MYJ (Mellor-Yamada-Janjic) scheme for the PBL (planetary boundary layer), the Goddard shortwave radiation scheme and the RRTM (rapid radiative transfer model) longwave radiation scheme, the Eta similarity scheme for surface layer formulation and the Noah LSM (Land Surface Model) to parameterize physics of land surface. A few preliminary tests are performed to assess the best cumulus parameterization scheme to be used both for the coarse and finest resolution domain for this event. Hence, the following parameterizations are tested: the new Kain–Fritsch and the Grell 3D schemes. The latter is an enhanced Dudhia of the Grell-Deveneyi scheme, in our simulations only used on the lowest resolution domain, where the option *cugd_avedx* (subsidence spreading) is switched on. Based on the results of these two cumulus parameterization schemes, the one producing the best precipitation forecast will be used to evaluate the impact of data assimilation.

**3 Data and methodology**

This section will be focused on the description of types of observations ingested into the assimilation procedure, namely both conventional and radar, and on the 3D-Var methodology as well as the observation operator used for the calculation of the reflectivity. Moreover, a brief overview of the evaluation method adopted to assess the performance of numerical weather predictions will be given.

**3.1 Observations to be assimilated**

Conventional observations SYNOP and TEMP were retrieved from the ECMWF Meteorological Archival and Retrieval System (MARS). They have been packed in a suitable format for ingest into the assimilation procedure using the Observation Preprocessor (OBSPROC) module provided by the 3D-Var system. Among its main functions there are also to perform a quality control check and to assign observational errors based on a pre-specified error file. In short, a total of 983 observations (967 SYNOP and 16 TEMP) are ingested into the coarse resolution domain, whereas a total of 338 (333 SYNOP and 5 TEMP) observations into the high resolution one.

Reflectivities taken from three C-band Doppler radars operational during the IOP4 have been assimilated to improve IC. The radars have different technical characteristics and were operated with different scanning strategies and operational settings as shown in Table 1: each radar has a half power beam width of 1.6, 1 and 0.9 degree respectively for Monte Midia (MM), Polar55C (P55C) and San Pietro Capofiume (SPC) and a range resolution of 500, 75 and 250 metres.

MM and SPC radars are included in the Italian weather radar network, while P55C radar is a research radar working on demand, but was operational during the IOPs of the HyMeX campaign (Roberto et al., 2016).

It is worth mentioning that radar data can be affected by numerous sources of errors, mainly due to ground clutter, attenuation due to propagation or beam blocking, anomalous propagation and radio interferences. This is the reason why a preliminary "cleaning" procedure is applied to the measured radar reflectivity from the three radars before the assimilation process, consisting of the following 3 steps:

- a first quality check of radar volumes to filter out radar pixels affected by ground clutter and anomalous propagation. Furthermore, Z was corrected for attenuation using a methodology based on the specific differential phase shift ($K_{dp}$) available for dual polarization radars (Vulpiani et al, 2015); moreover, reflectivity

is not corrected for partial beam blocking: all the data that are affected by partial beam blocking and clutter

have been filtered out;

•   volume reflectivity radar data are converted from their native polar coordinates (range, azimuth and elevation)

into geographical Cartesian ones (latitude, longitude and elevation);

•   the minimum assimilated reflectivity is set to -20 dBZ.

Moreover, no observation thinning is performed because this procedure is not yet developed into the 3D-Var system for

radar data. Instead, an iterative approach has been applied to extract more information from radar data during the

assimilation procedure: this is the multiple outer loops technique explained later in Section 4.

**3.2 3D-Var data assimilation method**

Data assimilation (DA) is a technique employed in many fields of geosciences perhaps most importantly in weather

forecasting and hydrology. In this context it is the procedure by which observations are combined with the product (*first*

*guess* or *background forecast*) of a NWP model and their corresponding error statistics, to produce a bettered estimate

(the *analysis*) of the true state of the atmosphere (Skamarock et al., 2008). The variational DA method realizes this

through the iterative minimization of a penalty function (Ide et al., 1997):

$J(\boldsymbol{x}) = J^b(\boldsymbol{x}) + J^0(\boldsymbol{x}) = \frac{1}{2}\{[\boldsymbol{y}^0 - H(\boldsymbol{x})]^T \boldsymbol{R}^{-1}[\boldsymbol{y}^0 - H(\boldsymbol{x})] + (\boldsymbol{x} - \boldsymbol{x}^b)^T \boldsymbol{B}^{-1}(\boldsymbol{x} - \boldsymbol{x}^b)\},$        (1)

where $\boldsymbol{x}^b$ is the first guess state vector, $\boldsymbol{y}^0$ is the assimilated observation vector, $H$ is the observation operator that links

the model variables to the observation variables and $\boldsymbol{x}$ is the unknown analysis state vector to be found by minimizing

*J(x)*. Finally, $\boldsymbol{B}$ and $\boldsymbol{R}$ are the background covariance error matrix and the observation covariance error matrix,

respectively.

The minimization of the penalty function *J(x)*, displayed by Equation (1), is the a posteriori maximum likelihood

estimate of the true atmosphere state, given the two sources of a priori data that are $\boldsymbol{x}^b$ and $\boldsymbol{y}^0$ (Lorenc, 1986).

In this study the 3D-Var system developed by Barker et al. (2003, 2004) is used for assimilating radar reflectivity and

conventional observations SYNOP and TEMP. The penalty function minimization is performed in a preconditioned

control variable space, where the preconditioned control variables are pseudo relative humidity, stream function,

unbalanced temperature, unbalanced potential velocity and unbalanced surface pressure. Because of radar reflectivity

assimilation is considered, the total water mixing ratio $q_t$ is chosen as the moisture control variable. The following

equation presents the observation operator used by the 3D-Var to calculate reflectivity for the comparison with the

observed one (Sun and Crook, 1997):

$Z = 43.1 + 17.5 \, log(\rho q_r),$        (2)

where $\rho$ and $q_r$ are the air density in kg/m$^3$ and the rainwater mixing ratio in g/kg, respectively, while $Z$ is the co-polar

radar reflectivity factor expressed in dBZ. Since the total water mixing ratio $q_t$ is used as the control variable, a warm

rain process (Dudhia, 1989) is introduced into the WRF-3D-Var system to allow for producing the increments of moist

variables linked to the hydrometeors.

The performance of the DA system strongly depends on the quality of the $B$ matrix in Equation (1). In this study, a
specific background error statistics is computed for both domains for the entire SOP1 duration using the National
Meteorological Center (NMC) method (Parrish and Derber, 1992). This technique estimates the initial state error using
differences of couples of forecasts valid at the same time, but with one of them having a delayed start time. One of the
advantage of this method is that it maintains information on the dynamic of the model itself, but it may not give the
proper correlation structure on data-sparse observations. Commonly, for regional applications and to remove the diurnal
cycle, a delay of 24 hours between the forecasts (T+24 minus T+12) is used; nevertheless, this delay can produce
overestimated correlation length scales compared to those needed by a variational data assimilation technique, because
of too dynamically evolved structures (Sadiki et al., 2000). Since 3D-Var is applied to the Mediterranean area, $B$ has to
take into account the scale of the motions of this orographic and meteorologically complex area: the model grid
resolution ranges between 12 km and 3km, therefore the errors have to describe the physical phenomena relative to
these scales.

**3.3 Evaluation**

The Point-Stat Tool of MET (Model Evaluation Tools) application (DTC, 2013), developed at the DTC (Developmental
Testbed Center, NCAR), has been used to objectively evaluate the 12 hours accumulated precipitation produced by
WRF on both domains. The interpolation method used to match the gridded model output to the point observation is the
distance weighted mean in a 3 x 3 square of grid points. The observations used for the statistical evaluation were
obtained from the DEWETRA platform of the Department of Civil Protection and the comparison has been performed
over central Italy target area using about 3000 rain gauges with a good coverage throughout the Italian territory.
Moreover, for interpreting results from the verification analysis bootstrap, confidence intervals (CIs) have been used to
analyze the uncertainty associated with the score's values. Bootstrapping is a non parametric, computationally
expensive, statistical technique (Efron & Tibshirani, 1993) for estimating parameters and uncertainty information, that
allows to make inferences from data without making strong distributional assumptions about the data or the statistic
being calculated. Therefore, the idea was to estimate CIs to set some bounds (bootstrap upper and lower confidence
limits) on the expected value of the verification score helping to assess whether differences between competing
forecasts are significant.

**4 Design of the numerical experiments: discussion of the results**

The simulations on the coarser resolution domain (D01) are run from 12:00UTC of 13 September 2012 and integrated
for the following 96 hours, whereas runs on the finest resolution domain started at 00:00UTC of September 14 for a
total of 48 hours of integration. The previous coarser resolution WRF forecast at 00:00UTC is used as the first guess in
the 3D-Var experiment, because 00:00UTC has been selected as the "*analysis time*" of the assimilation procedure. After
assimilation, the lateral and lower boundary conditions are updated for the high resolution forecast. Finally, the new IC
and BC are used for the model initialization (in a warm start regime) at 00:00UTC. As already pointed out a set of
preliminary experiments are performed using different cumulus convective scheme to assess the best one to be used.
The following experiments are performed without assimilation and using the convective scheme on the coarser
resolution domain only: KAIN-FRITSCH (KF_MYJ); GRELL3D (GRELL3D_MYJ); GRELL3D associated with the
CUGD factor (GRELL3D_MYJ_CUGD). The best performance is obtained by Grell3D scheme which is able to
simulate the peak precipitation cumulated in 24h over Campo Imperatore, whereas KAIN-FRITSCH completely misses
it (not shown here). The MET statistical analysis support the previous finding and the simulation with *cugd_avedx*
activated shows a significant performance in terms of uncertainty of the calculated scores than the other two simulations
(not shown). Here after GRELL3D_MYJ_CUGD is referred as the control experiment (CTL) performed without any
data assimilation.
At this point analysis of a new set of simulations is performed allowing to establish the best model configuration for the
radar reflectivity assimilation. The DA experiments aim to investigate:
1.  the impact of the assimilation at low and high resolution by assimilating both conventional and non-
conventional data at both resolutions;
2.  the impact of the assimilation of different types of observations;
3.  the impact of the different radars, which is investigated by performing experiment by assimilating conventional
data and then adding radar one by one.

Therefore in Table 2, together with CTL simulation, the following DA experiments are summarized: i) the assimilation
of conventional data only (CON); ii) the assimilation of reflectivity data from MM only (CONMM) are added; iii) the
assimilation of P55C radar reflectivity is added to the previous experiments (CONMMPOL); iv) the assimilation of the
third radar reflectivity data is added to the previous (CONMMPOLSPC). Finally, an experiment to assess the role of the
outer loops is performed (CONMMPOLSPC3OL): to include non-linearities into the observation operator and to
evaluate the impact of reflectivity data entering for each cycle, the multiple outer loops strategy is applied (Hsiao et al.
2012). According to this approach, the non-linear problem is solved iteratively as a progression of linear problems: the
assimilation system is able to ingest more observations by running more than one analysis outer loop, allowing
observations rejected in the previous loop to be enter into the subsequent one. Since radar data are non linearly related
to the analysis control variables, the outer loops method is particularly helpful to extract more information from such
data.
In the following section the results will be presented and discussed following the rationale of the previously introduced
experiments and analyzing the uncertainty (confidence level of 95%) in the realized scores (Forecast Accuracy (ACC),
Frequency Bias (FBIAS), Equitable Threat Score (ETS), False Alarm Ratio (FAR)) for performance quantitative
assessment.
**5 Impact of conventional measurements and radar reflectivity assimilation on rainfall forecast: low versus high**
**resolution**
In figure 6, a preliminary comparison among low resolution (LR) simulations is shown. The control simulation (CTL)
without data assimilation is shown in Figure 6a; whereas the other panels (b, c, d, e, f) show the experiments performed
using the data assimilation.
The outputs of different experiments in Fig. 6 have been eyeballed and we found that CONMMPOLSPC_LR_12KM
(black arrow in Fig. 6e) shows the most encouraging performance compared to the observed accumulated rainfall of
Figure 2: the rainfall maximum over Campo Imperatore is very well simulated, however a slight cell displacement at the

border between Marche and Abruzzo regions is noticeable. The rain cumulated by the gauges in 24h related to this cell is around 300 mm (Fig. 3c); in the simulations shown in Figures 6b and 6f this cell is reproduced, although its position is shifted in another region. Furthermore, the precipitation pattern along the northern coasts of Abruzzo (black oval in Fig. 6e) is also quite well forecasted. At an objective comparison of the statistical indices (not shown here) with their relative upper and lower confidence limits for the 12 hours accumulated precipitation and for two thresholds (1 mm and 40 mm for light and heavy rain regimes respectively), we obtained likely good values for ACC and FAR for all the experiments and for heavy rain regimes, strengthened by a small uncertainty interval. On the other hand, for the lower threshold the values of FBIAS for all simulations, considering also the confidence intervals, are greater than one. One possible interpretation of the impact of the lower threshold is that with 95% confidence all the experiments are overestimating the frequency of precipitation around 1 mm/12h.

Similarly to the above comparison, in figure 7 high resolution results (HR) obtained performing reflectivity assimilation on 12 km domain (column 1), on 3 km (column 2) and on 12 km and 3 km together (column 3) are presented; to the top of figure 7 the CTL experiment on D02 is shown. Figure 7 is organized as follows: viewing panels by line, on line 1 all the simulations with conventional data assimilation only (CON*) are found; on line 2 all the experiments with the assimilation of the reflectivity data from MM radar added (CONMM*); on line 3 all the experiments with the assimilation of the reflectivity data from 2 C-band radars added (CONMMPOL*); on line 4 all the experiments with the assimilation of the reflectivity data from all 3 C-band radars added (CONMMPOLSPC*); on line 5 the simulations where the strategy of outer loops is adopted (CONMMPOLSPC3OL*). In order to quantify the uncertainty associated to these experiments, the bootstrap 95% confidence intervals for verification statistics ACC, FBIAS, ETS, FAR have been summarized over tables (from 3 to 9) reporting the two thresholds of precipitation for light and heavy rain regimes: 1 mm/12h and 40 mm/12h, respectively.

In order to investigate the impact of the assimilation at different resolutions, we examine figure 7 by column and comparing it with the available observations (Fig. 2) using also the statistical analysis:

- column 1 (12KM): CTL produces an overestimation of the rainfall that is not corrected by the assimilation of conventional data, but assimilating the reflectivity from the 3 radars (column 1 line 4) and also introducing the 3 outer loops (column 1 line 5) the main cells are better reproduced. MET indices (not shown here) suggest that CTL and CON_HR_12KM have the largest difference between the CIs bounds for higher thresholds of FBIAS: this result suggests that the remaining simulations, with smallest difference in CIs limits and with both bounds lower than 1, surely underestimate the frequency of heavy precipitating events. Another aspect to point out is that some indices for all simulations are quite close to each other and within the CIs, so it is not possible to discern which is the best experiment over all;
- column 2 (3KM): a partial correction of the rainfall overestimation compared to column 1 is observed especially if reflectivity from all the radars are assimilated together with conventional data and the outer loops strategy is applied (column 2 line 5); the statistical indices in Table 3 show as the most competitive experiment among the assimilated ones the CONMMPOLSPC3OL_3KM for lower threshold of rain for ACC (0.83) and FBIAS (0.96), on the other hand CONMM_3KM is the most promising simulation for heavy rain threshold for the indices FBIAS (0.31) and ETS (0.13);
- column 3 (12KM_3KM): rainfall overestimation was partially corrected compared to columns 1 and 2 by all the experiments; the MET statistics in Table 4 shows that CTL and CONMMPOLSPC3OL_12KM_3KM are

 Furthermore, general improvements (especially for FBIAS and ETS) come out for heavy rain regimes when radar reflectivity assimilation has been performed on the highest resolution domain, whereas the ingestion of conventional observations produces the worst results for FBIAS and ETS since a smaller number of them were assimilated into the finest resolution domain (for instance one sounding on five total) than that the coarser one. Data assimilation, operated on both 12 km and 3 km, shows similar performances to the experiments where assimilation is performed only on D01 (table 4), but a worse response for higher thresholds (tables 3 and 4) than the ones where assimilation is carried out on D02.

In order to examine the impact of the assimilation of different data and radars, we can now analyze the experiments showed in figure 7 line by line. The results are compared with the observations of Fig. 2. The following considerations are worth discussing:

- line 1 (CON): a strong reduction of the rainfall is found with respect to CTL if conventional data are assimilated, but the rainfall pattern remains unchanged. Statistical indices of CON experiment (Table 5) do not improve the performances of CTL (despite a reduction in some cases of the spread between the CIs limits for higher thresholds of the FBIAS). Some indices values suggest a slightly better performance when the conventional observations are assimilated only on the bigger domain and for higher thresholds (FBIAS 0.49), together with an improvement of FAR index for heavy rain regime (FAR 0.001);

- line 2 (CONMM): a further reduction in the precipitation overestimation is found as well as some variations in the pattern of the rainfall; the scores in Table 6, together with their bootstrap upper and lower limits, show that MM radar reflectivity and conventional observations assimilation, improves the model performance above all for lower thresholds respect to the experiments where only SYNOP and TEMP were ingested (comparing scores of Table 6 with ones of Table 5). It applies also for some of the scores at higher thresholds (for example for ETS);

- line 3 (CONMMPOL): a quite strong improvement in the rainfall amount is found for all simulations. However, from the statistics of Table 7, compared to the one in Table 6, we found a general worsening of the results both for light and heavy rain regimes when POL is added (especially for FBIAS and ETS, in some cases also for ACC and FAR at lower thresholds);

- line 4 (CONMMPOLSPC): a clear correction of the rainfall pattern is found; the overestimation produced by the simulation where the reflectivity from all the radars are assimilated on the 3 km domain has been corrected by the experiment in which the reflectivity is assimilated both on D01 and D02; the uncertainty in the realized scores of Table 8 suggests that the addition of SPC radar improves the results, furthermore they are not better than those where only MM is ingested;

- line 5 (CONMMPOLSPC3OL): the outer loops experiment confirms the strong overestimation reduction by *12KM_3KM; from Table 9 it seems that the introduction of 3OL improves the indices estimate and bounds

above all when the 12 km domain is considered (see FBIAS and ETS for both rain regimes and FAR for lower
thresholds).
In summary, simulations results show that assimilation of conventional data is better to perform on the lowest resolution
domain because more observations were used in the coarser domain, whereas when the assimilation is performed on the
highest resolution domain only few SYNOP and even less TEMP fell down in the 3 km domain at the analysis time of
the assimilation procedure. The impact of the conventional observations are expected to be lower than those of the non
conventional ones, because most of them have already been used by ECMWF to produce their analysis and that they are
here used as first guess, even if at lower resolution (0.25°). Therefore, they result to be correlated to the background and
the improvements of those experiments where they are assimilated are expected to be low.
With regard to the assimilation of reflectivity radar data, it should be noted that P55C radar observations of the event
considered is shielded at the lowest elevation angles by the Apennines range and provides a limited contribution to
reflectivity data that are assimilated. Also the outer loops strategy could have an important role in the assimilation
procedure, but this latter needs a further investigation (for example an additional work has to be dedicated to testing the
different tuning factors for both observation and background during each outer loop) because a general rainfall
underestimation for higher thresholds is found.
The results of this section confirm that when there is a correlation between the observations and the first guess used, the
results of the data assimilation are poor, especially if no "special" observation is available on a wide area. The
assimilation of a large amount of surface data together with the radiosonde ones decreases the quality of the final
analysis produced. It probably depends on the different density of the surface and the three dimensional data of
radiosondes, as assessed by Liu and Rabier (2002), being the former much larger than the latter.

**6 Conclusions**
In this manuscript the effects of multiple radar reflectivity data assimilation on a heavy precipitation event occurred
during the SOP1 of the HyMeX campaign have been evaluated: the aim is to build a regionally-tuned numerical
prediction model and decision-support system for environmental civil protection services within the central Italian
regions. A sensitivity study at different domain resolution and using different types of data to improve initial conditions
has been performed by assimilating into the WRF model radar reflectivity measurements, collected by three C-band
Doppler weather radars operational during the event that hit central Italy on 14 September 2012. The 3D-Var and MET
are the WRF tools used to assess this purpose. The study is performed on the complex basin, both for the orography and
physical phenomena, of the Mediterranean area. First of all, WRF model responses to different types of cumulus
parameterizations have been tested to establish the best configuration and to obtain the control simulation. The latter has
been compared with observations and other experiments performed using 3D-Var. The set of assimilation experiments
have been conducted following two different strategies: i) data assimilation at low and high resolution or at both
resolutions simultaneously; ii) conventional data against radar reflectivity data assimilation. Both have been examined
to assess the impact on rainfall forecast.
The major findings of this work have been the following:

- Grell 3D parameterization improves the simulations both on D01and D02 and the use of the spreading factor is an added value in properly predict heavy rainfall over inland of Abruzzo and the rainfall pattern along the northeast coast;

- investigating the impact of the assimilation at different resolutions, positive results are showed by the experiments where the data assimilation is performed on both domains 12 km and 3 km;

- the impact of the assimilation using different types of observations shows improvements if reflectivity from all the radars, along with SYNOP and TEMP are assimilated; furthermore, MM is the one that gives more optimistic results due to its excellent monitoring of the whole event;

- the outer loops strategy allows for further improving positive impact of the assimilation of multiple reflectivity radars data. Moreover, a deeper investigation of this approach is required to well assess its impact, above all concerning the running time in an operational context;

- we have seen that there are thresholds where the WRF 3D-Var is statistically significant, with 95% confidence, while for other thresholds we have to be careful in drawing conclusions above all in the face of large uncertainty or when the score values are quite close to each other.

From the results obtained in this study, it is not possible to assess, in general terms, which is the best model configuration. In fact, this analysis should be performed systematically with a significant number of flash flood case studies before one can claim with certainty the positive impact of multiple reflectivity radar observations assimilation upon the forecast skill. Nevertheless, this work has pointed out aspects in 3D-Var reflectivity data assimilation that encourages to investigate more flash flood events occurred over central Italy, in order to make the proposed approach suitable to provide a realistic prediction of possible flash floods both for the timing and localization of such events. To confirm and consolidate these initial findings, apart from analyzing more case studies, a deeper analysis of the meteorology of the region and of the performance of the data assimilation system throughout longer trials in a "pseudo-operational" procedure is necessary. Moreover, a more sophisticated spatial verification technique (MODE, Method for Object-Based Diagnostic Evaluation, Davis et al., 2006a, 2006b) which focuses on the realism of the forecast, by comparing features or 'objects' that characterize both forecast and observation fields, could be investigated in the future. In fact, spatial verification methods are particularly suitable to address the model capability to reproduce structures like the convective systems responsible for the high precipitation events as considered in the present research, which, because of their typical dimensions, need high resolution simulations to be predicted (Gilleland et al., 2009). These new-generation spatial verification methods, through the identification and the geometrical description of 'objects' in forecast and observation fields (e.g. accumulated precipitation or radar reflectivity), permit an evaluation of the forecast skill in a more consistent way.

**Acknowledgements**

We are grateful to the Gran Sasso National Laboratories for support in computing resources, as well as the National Civil Protection Department and CIMA Research Foundation for rain gauges data using for the model validation. NCAR is also acknowledged for WRF model, 3D-Var system and MET tool. This work aims at contributing to the HyMeX programme.

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

**LIST OF FIGURES**

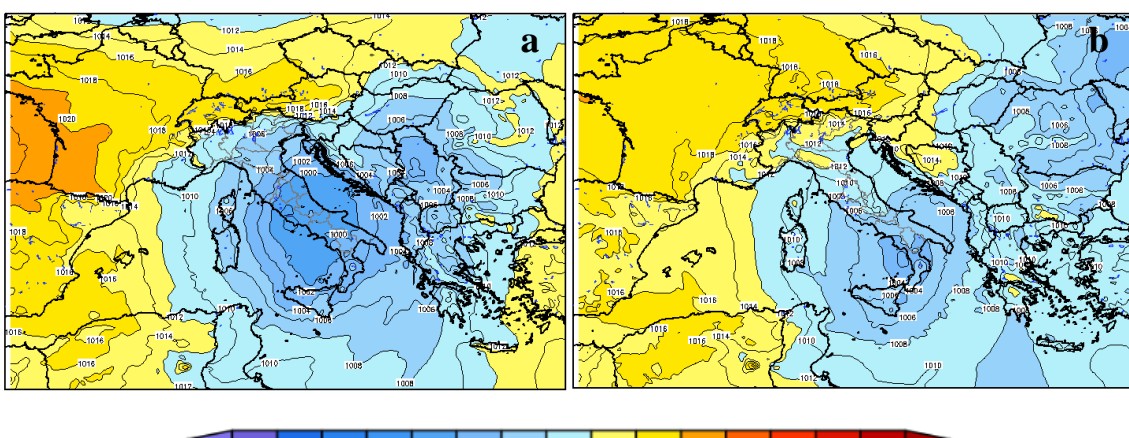


Mean sea level pressure [hPa]

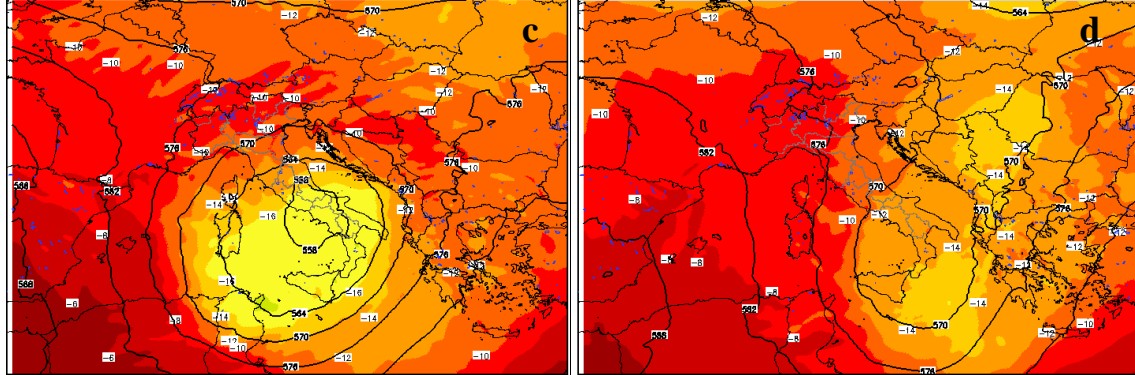


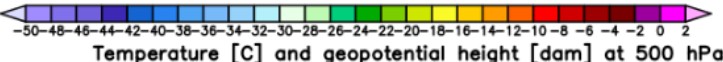


**Figure 1. ECMWF (European Center for Medium-Range Weather Forecasts) analyses at 12:00UTC on 14**
**September 2012: a) mean sea level pressure, c) temperature (color shades) and geopotential height (black isolines) at**
**500 hPa; ECMWF analyses at 12:00UTC on 15 September 2012: b) mean sea level pressure, d) temperature (black**
**isolines) and geopotential height (color shades) at 500 hPa.**


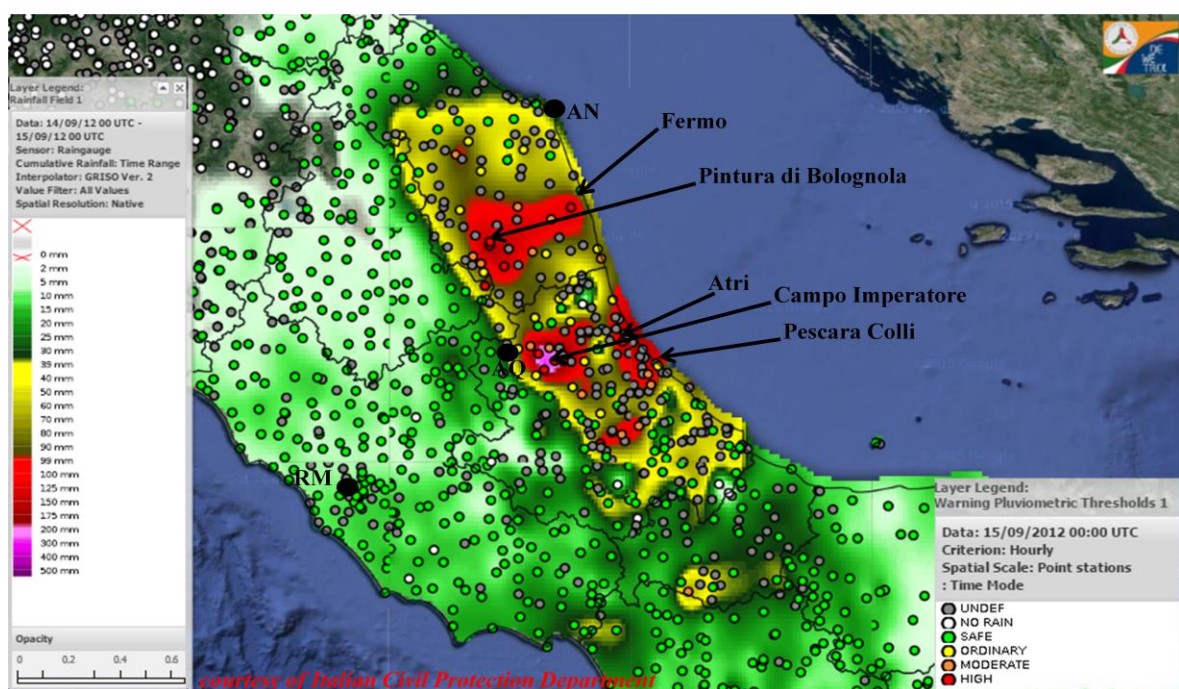

**Figure 2: Interpolated map of 24h accumulated rainfall from 00:00UTC of 14 September 2012 over Abruzzo and Marche**
**regions taken from DEWETRA system from rain gauges measurements.**
**Black contours are the administrative boundaries of regions, while the colored circles represent the warning pluviometric**
**thresholds.**

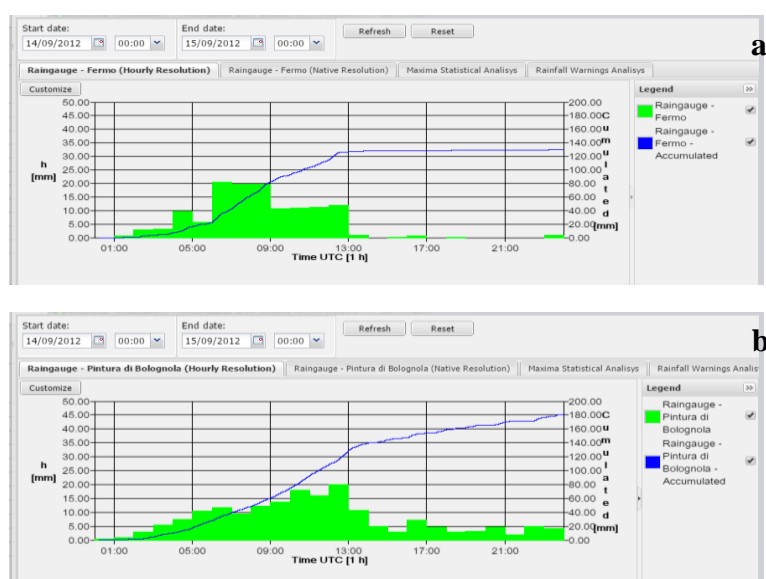




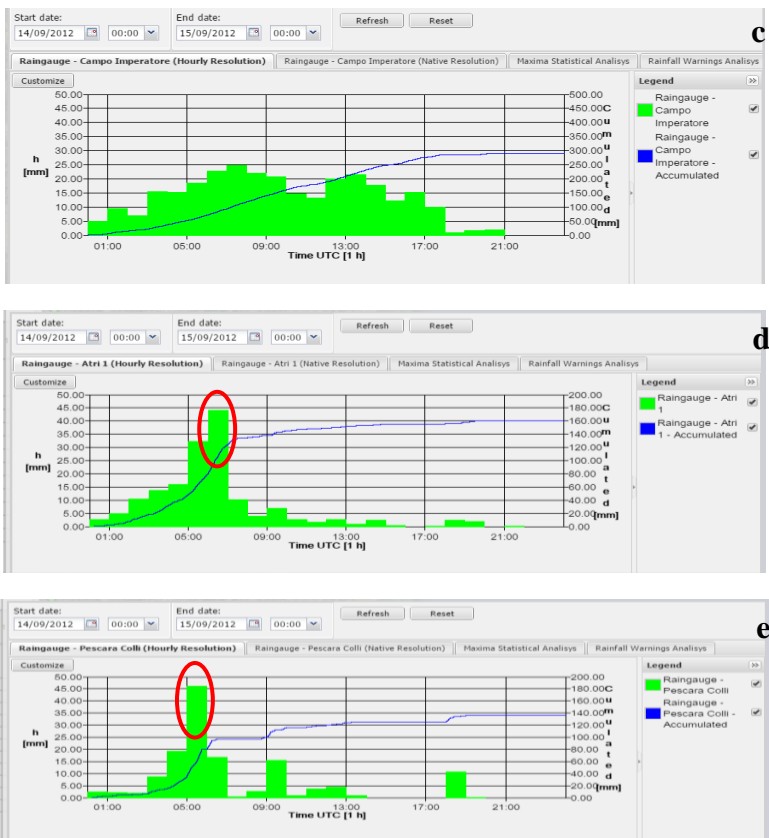



Figure 3: Rain gauges time series of some selected stations in Marche (a, Fermo and b, Pintura di Bolognola) and Abruzzo (c, Campo Imperatore, d, Atri and e, Pescara Colli) regions during the event of 14 September 2012. The green histogram represents the hourly accumulated precipitation (scale on the left); the blue line represents the incremental accumulation within the 24h (scale on the right). (*courtesy of Italian Civil Protection Department*)


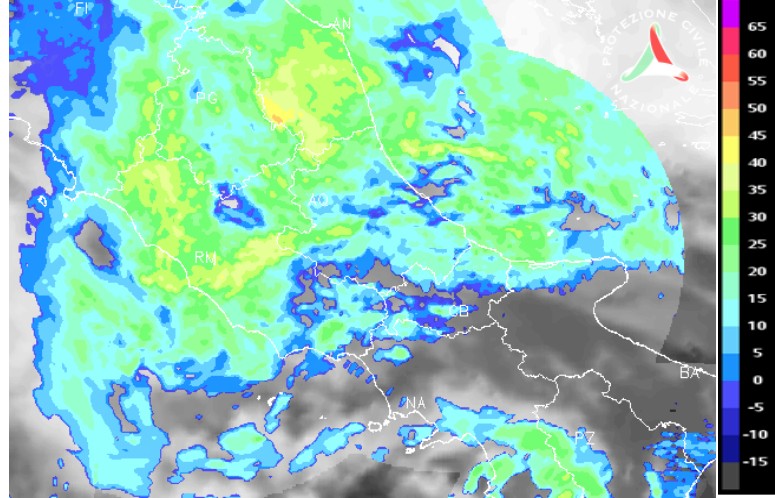


Figure 4: Zoom over central Italy of the reflectivity on 14 September 2012 at 08:00UTC from the Italian radar network overlapped with the MSG (IR 10.8) at 07:30UTC. (*courtesy of Italian DPC*)


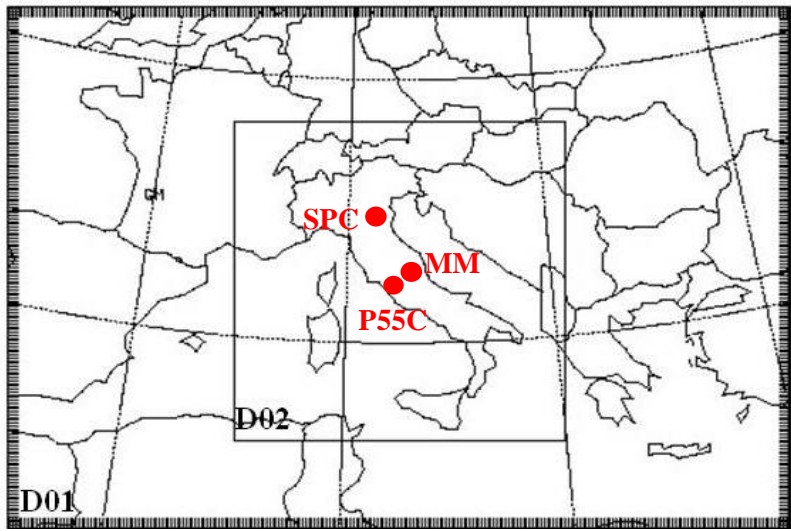


**Figure 5: WRF *ndown* domains configuration: the two domains have respectively resolution of 12km and 3km. The high**
**resolution D02 over Italy includes Mt. Midia (MM), ISAC-CNR (P55C) and San Pietro Capofiume (SPC) radars (red dots in**
**the figure).**




**Figure 6: WRF D01 accumulated 24h rainfall forecast over central Italy from 00:00UTC of 14 September 2012: a) WRF D01 CTL; b) WRF D01 CON_LR_12KM; c) WRF D01 CONMM_LR_12KM;d)WRF D01 CONMMPOL_LR_12KM; e) WRF D01 CONMMPOLSPC_LR_12KM; f) WRF D01 CONMMPOLSPC3OL_LR_12KM.**



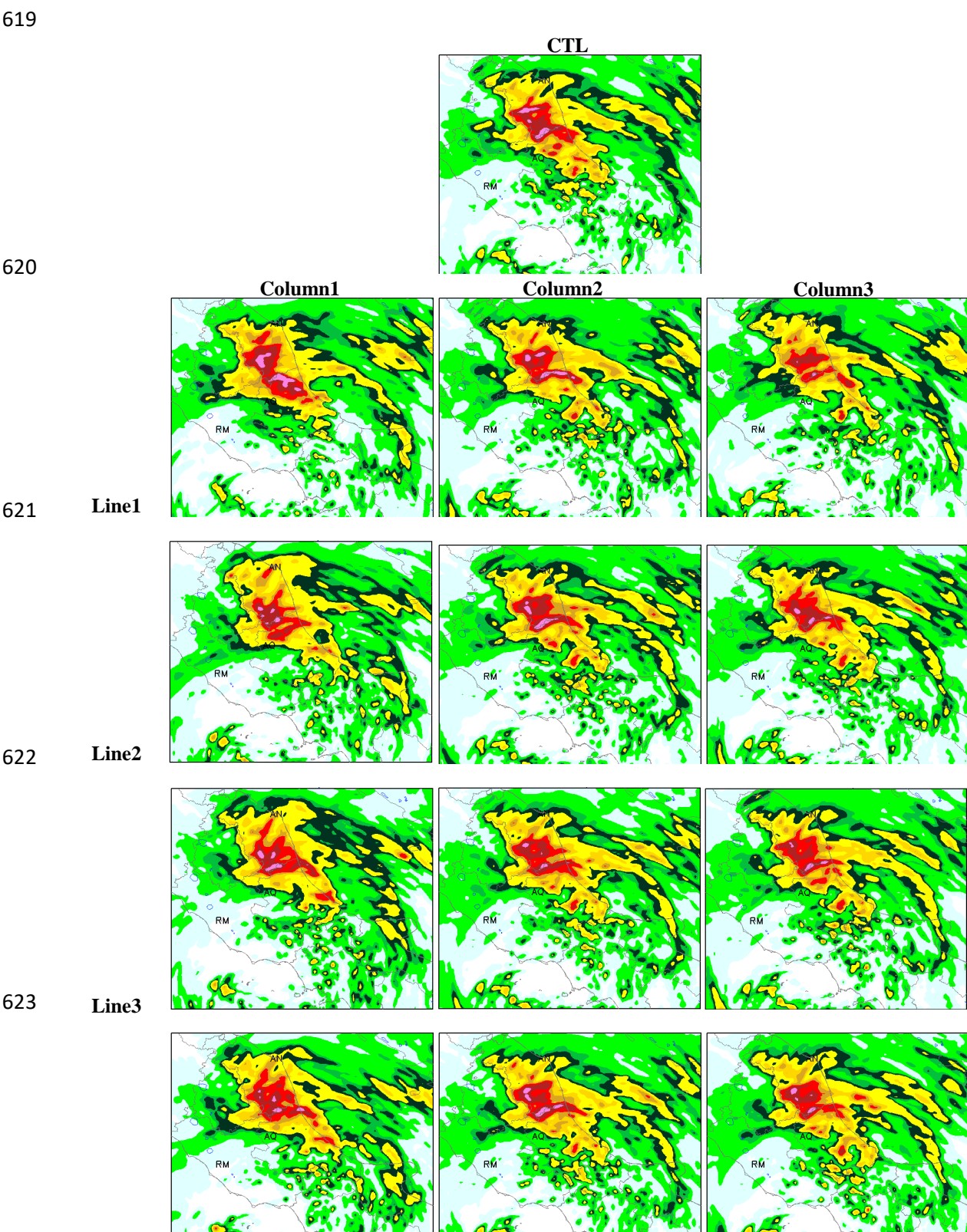


**Line1**
**Line2**
**Line3**
**Line4**

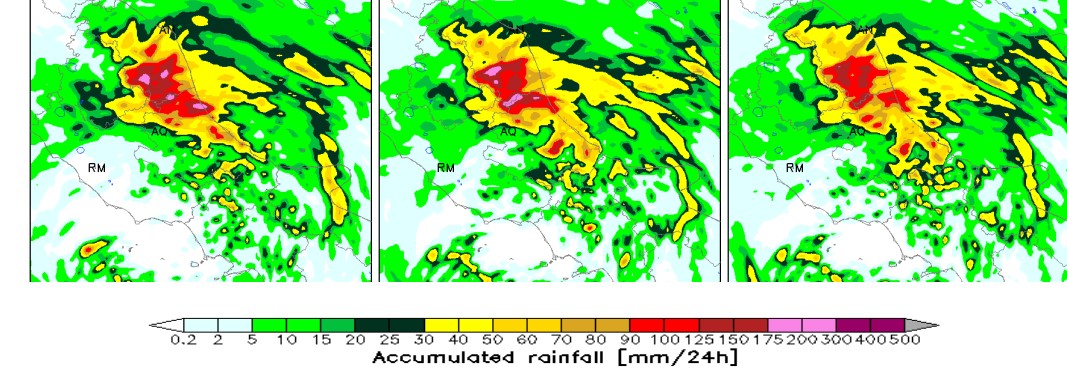

**Line5**

**Figure 7: WRF D02 accumulated 24h rainfall forecast over central Italy from 00:00UTC of 14 September 2012: CTL**
**simulation (top center); on each column simulations obtained performing reflectivity assimilation at different resolutions**
**(*12KM, *3KM, *12KM_3KM); on each line simulations performed assimilating different kinds of data (CON*, CONMM*,**
**CONMMPOL*,CONMMPOLSPC*, CONMMPOLSPC3OL*).**

**Table 1: Technical characteristics of the three radars whose reflectivity have been assimilated during IOP4.**

| Features | Units | MM radar | P55C radar | SPC radar |
|---|---|---|---|---|
| Owner | | CF Abruzzo Region | ISAC-CNR of Rome | Arpae Emilia Romagna |
| Location | | Monte Midia | Rome | San Pietro Capofiume |
| Latitude | [deg] | 42.057 | 41.840 | 44.6547 |
| Longitude | [deg] | 13.177 | 12.647 | 11.6236 |
| Height (a.s.l.) | [m] | 1760 | 131 | 31 |
| Doppler | | YES | YES | YES |
| Dual Polarization | | NO | YES | YES |
| Range Resolution | [m] | 500 | 75 | 250 |
| Half Power Beam Width | [deg] | 1.6 | 1 | 0.9 |
| Temporal Resolution | [min] | 15 | 5 | 15 |
| Elevations angles used in PPI scans | [deg] | 0, 1, 2, 3 | 0.6, 1.6, 2.6, 4.4, 6.2, 8.3, 11.0, 14.6 | 0.53, 1.4, 2.3, 3.2, 4.1, 5.0 |
| Maximum Range | [km] | 120 or 240 | 120 | 125 |


**Table 2: List of experiments to test the impact of data assimilation.**

| Experiment | Cumulus | Grid Resolution | Assimilation Synop+Temp | Assimilation Radar |
|---|---|---|---|---|
| CTL | GRELL3D+CUGD | 12KM/3KM | NO | NO |
| CON | GRELL3D+CUGD | 12KM/3KM/BOTH | YES | NO |
| CONMM | GRELL3D+CUGD | 12KM/3KM/BOTH | YES | MM |
| CONMMPOL | GRELL3D+CUGD | 12KM/3KM/BOTH | YES | MM+POL |

| CONMMPOLSPC | GRELL3D+CUGD | 12KM/3KM/BOTH | YES | MM+POL+SPC |
| CONMMPOLSPC3OL | GRELL3D+CUGD | 12KM/3KM/BOTH | YES | MM+POL+SPC with 3 outer loops |


**Table 3**: Bootstrap 95% confidence intervals for verification statistics Forecast Accuracy (ACC), Frequency Bias (FBIAS),
Equitable Threat Score (ETS), False Alarm Ratio (FAR) and referred to experiments in column 2. They are considered as a
function of thresholds (1mm/12h and 40mm/12h). The experiments are: CTL, CON_3KM, CONMM_3KM,
CONMMPOL_3KM, CONMMPOLSPC_3KM, CONMMPOLSPC3OL_3KM.

| Experiment | ACC Thresholds mm/12h | | FBIAS Thresholds mm/12h | | ETS Thresholds mm/12h | | FAR Thresholds mm/12h | |
|---|---|---|---|---|---|---|---|---|
| | 1 | 40 | 1 | 40 | 1 | 40 | 1 | 40 |
| CTL | (0.80) **0.83** (0.87) | (0.96) **0.98** (0.99) | (0.79) **0.94** (1.13) | (0.14) **0.47** (1.61) | (0.23) **0.33** (0.45) | (0.04) **0.10** (0.16) | (0.16) **0.21** (0.27) | (0.001) **0.007** (0.015) |
| CON_3KM | (0.78) **0.82** (0.85) | (0.96) **0.98** (0.99) | (0.65) **0.80** (0.98) | (0.08) **0.18** (0.42) | (0.14) **0.24** (0.35) | (0.03) **0.06** (0.12) | (0.17) **0.22** (0.28) | (0.001) **0.004** (0.009) |
| CONMM_3KM | (0.78) **0.82** (0.86) | (0.97) **0.98** (0.99) | (0.79) **0.96** (1.17) | (0.14) **0.31** (0.68) | (0.17) **0.26** (0.37) | (0.05) **0.13** (0.26) | (0.18) **0.24** (0.29) | (0.001) **0.005** (0.11) |
| CONMMPOL_3KM | (0.77) **0.81** (0.85) | (0.96) **0.98** (0.99) | (0.76) **0.94** (1.16) | (0.12) **0.28** (0.65) | (0.13) **0.23** (0.33) | (0.03) **0.09** (0.14) | (0.18) **0.24** (0.30) | (0.001) **0.006** (0.11) |
| CONMMPOLSPC_3KM | (0.78) **0.82** (0.86) | (0.96) **0.98** (0.99) | (0.85) **1.03** (1.25) | (0.10) **0.27** (0.83) | (0.18) **0.28** (0.39) | (0.03) **0.07** (0.13) | (0.19) **0.24** (0.31) | (0.001) **0.005** (0.012) |
| CONMMPOLSPC3OL_3KM | (0.79) **0.83** (0.86) | (0.97) **0.98** (0.99) | (0.81) **0.96** (1.17) | (0.10) **0.24** (0.64) | (0.17) **0.27** (0.39) | (0.05) **0.12** (0.19) | (0.21) **0.27** (0.33) | (0.000) **0.003** (0.007) |


**Table 4**: Bootstrap 95% confidence intervals for verification statistics Forecast Accuracy (ACC), Frequency Bias (FBIAS),
Equitable Threat Score (ETS), False Alarm Ratio (FAR) and referred to experiments in column 3. They are considered as a
function of thresholds (1mm/12h and 40mm/12h). The experiments are: CTL, CON_12KM_3KM, CONMM_12KM_3KM,
CONMMPOL_12KM_3KM, CONMMPOLSPC_12KM_3KM, CONMMPOLSPC3OL_12KM_3KM.

| Experiment | ACC Thresholds mm/12h | | FBIAS Thresholds mm/12h | | ETS Thresholds mm/12h | | FAR Thresholds mm/12h | |
|---|---|---|---|---|---|---|---|---|
| | 1 | 40 | 1 | 40 | 1 | 40 | 1 | 40 |
| CTL | (0.80) **0.83** (0.87) | (0.96) **0.98** (0.99) | (0.79) **0.94** (1.13) | (0.14) **0.47** (1.61) | (0.23) **0.33** (0.45) | (0.04) **0.10** (0.16) | (0.16) **0.21** (0.27) | (0.001) **0.007** (0.015) |

| | ACC | | FBIAS | | ETS | | FAR | |
|---|---|---|---|---|---|---|---|---|
| CON_**12KM_3KM** | (0.77) | (0.96) | (0.68) | (0.02) | (0.11) | (0.01) | (0.21) | (0) |
| | **0.81** | **0.98** | **0.84** | **0.10** | **0.20** | **0.04** | **0.27** | **0.001** |
| | (0.84) | (0.99) | (1.03) | (0.34) | (0.30) | (0.007) | (0.33) | (0.004) |
| CONMM_**12KM_3KM** | (0.79) | (0.96) | (0.79) | (0.09) | (0.18) | (0.03) | (0.17) | (0.001) |
| | **0.83** | **0.98** | **0.96** | **0.31** | **0.28** | **0.07** | **0.23** | **0.006** |
| | (0.86) | (0.99) | (1.18) | (1.02) | (0.40) | (0.13) | (0.29) | (0.013) |
| CONMMPOL_**12KM_3KM** | (0.77) | (0.96) | (0.79) | (0.11) | (0.14) | (0.03) | (0.19) | (0.001) |
| | **0.81** | **0.98** | **0.96** | **0.26** | **0.23** | **0.08** | **0.25** | **0.006** |
| | (0.85) | (0.99) | (1.19) | (0.65) | (0.33) | (0.14) | (0.31) | (0.011) |
| CONMMPOLSPC_**12KM_3KM** | (0.77) | (0.97) | (0.87) | (0.09) | (0.16) | (0.04) | (0.22) | (0) |
| | **0.81** | **0.98** | **1.04** | **0.25** | **0.26** | **0.08** | **0.28** | **0.004** |
| | (0.85) | (0.99) | (1.28) | (0.70) | (0.37) | (0.14) | (0.34) | (0.009) |
| CONMMPOLSPC3OL_**12KM_3KM** | (0.79) | (0.97) | (0.82) | (0.08) | (0.19) | (0.05) | (0.19) | (0) |
| | **0.83** | **0.98** | **0.98** | **0.15** | **0.30** | **0.11** | **0.25** | **0.002** |
| | (0.86) | (0.99) | (1.18) | (0.24) | (0.41) | (0.18) | (0.31) | (0.003) |


**Table 5**: Bootstrap 95% confidence intervals for verification statistics Forecast Accuracy (ACC), Frequency Bias (FBIAS),
Equitable Threat Score (ETS), False Alarm Ratio (FAR) and referred to experiments in line 1. They are considered as a
function of thresholds (1mm/12h and 40mm/12h). The experiments are: CTL, CON_3KM, CON_HR_12KM,
CON_12KM_3KM.

| | ACC | | FBIAS | | ETS | | FAR | |
|---|---|---|---|---|---|---|---|---|
| **Experiment** | *Thresholds mm/12h* | | *Thresholds mm/12h* | | *Thresholds mm/12h* | | *Thresholds mm/12h* | |
| | *1* | *40* | *1* | *40* | *1* | *40* | *1* | *40* |
| CTL | (0.80) | (0.96) | (0.79) | (0.14) | (0.23) | (0.04) | (0.16) | (0.001) |
| | **0.83** | **0.98** | **0.94** | **0.47** | **0.33** | **0.10** | **0.21** | **0.007** |
| | (0.87) | (0.99) | (1.13) | (1.61) | (0.45) | (0.16) | (0.27) | (0.014) |
| **CON**_3KM | (0.78) | (0.96) | (0.65) | (0.08) | (0.14) | (0.03) | (0.17) | (0.001) |
| | **0.82** | **0.98** | **0.80** | **0.18** | **0.24** | **0.06** | **0.22** | **0.004** |
| | (0.85) | (0.99) | (0.98) | (0.42) | (0.35) | (0.12) | (0.28) | (0.009) |
| **CON**_HR_12KM | (0.77) | (0.96) | (0.75) | (0.21) | (0.15) | (0.03) | (0.20) | (0.005) |
| | **0.81** | **0.97** | **0.91** | **0.49** | **0.25** | **0.07** | **0.26** | **0.0011** |
| | (0.85) | (0.99) | (1.11) | (1.61) | (0.36) | (0.13) | (0.31) | (0.19) |
| **CON**_12KM_3KM | (0.77) | (0.96) | (0.68) | (0.02) | (0.11) | (0.01) | (0.21) | (0) |
| | **0.81** | **0.98** | **0.84** | **0.10** | **0.20** | **0.04** | **0.27** | **0.001** |
| | (0.84) | (0.99) | (1.03) | (0.34) | (0.30) | (0.07) | (0.33) | (0.004) |


**Table 6**: Bootstrap 95% confidence intervals for verification statistics Forecast Accuracy (ACC), Frequency Bias (FBIAS),
Equitable Threat Score (ETS), False Alarm Ratio (FAR) and referred to experiments in line 2. They are considered as a
function of thresholds (1mm/12h and 40mm/12h). The experiments are: CTL, CONMM_3KM, CONMM_HR_12KM,
CONMM_12KM_3KM.

| Experiment | ACC Thresholds mm/12h | | FBIAS Thresholds mm/12h | | ETS Thresholds mm/12h | | FAR Thresholds mm/12h | |
|---|---|---|---|---|---|---|---|---|
| | 1 | 40 | 1 | 40 | 1 | 40 | 1 | 40 |
| CTL | (0.80) **0.83** (0.87) | (0.96) **0.98** (0.99) | (0.79) **0.94** (1.13) | (0.14) **0.47** (1.61) | (0.23) **0.33** (0.45) | (0.04) **0.10** (0.16) | (0.16) **0.21** (0.27) | (0.001) **0.007** (0.15) |
| **CONMM**_3KM | (0.78) **0.82** (0.86) | (0.97) **0.98** (0.99) | (0.79) **0.96** (1.17) | (0.14) **0.31** (0.68) | (0.17) **0.26** (0.37) | (0.05) **0.13** (0.26) | (0.18) **0.24** (0.29) | (0.001) **0.005** (0.011) |
| **CONMM**_HR_12KM | (07.8) **0.82** (0.86) | (0.97) **0.98** (0.99) | (0.79) **0.95** (1.16) | (0.15) **0.29** (0.64) | (0.18) **0.28** (0.39) | (0.07) **0.14** (0.21) | (0.19) **0.24** (0.31) | (0) **0.004** (0.008) |
| **CONMM**_12KM_3KM | (0.79) **0.83** (0.86) | (0.96) **0.98** (0.99) | (0.79) **0.96** (1.18) | (0.09) **0.31** (1.01) | (0.18) **0.28** (0.40) | (0.03) **0.07** (0.13) | (0.17) **0.23** (0.29) | (0.001) **0.006** (0.013) |


Table 7: Bootstrap 95% confidence intervals for verification statistics Forecast Accuracy (ACC), Frequency Bias (FBIAS),
Equitable Threat Score (ETS), False Alarm Ratio (FAR) and referred to experiments in line 3. They are considered as a
function of thresholds (1mm/12h and 40mm/12h). The experiments are: CTL, CONMMPOL_3KM,
CONMMPOL_HR_12KM, CONMMPOL_12KM_3KM.

| Experiment | ACC Thresholds mm/12h | | FBIAS Thresholds mm/12h | | ETS Thresholds mm/12h | | FAR Thresholds mm/12h | |
|---|---|---|---|---|---|---|---|---|
| | 1 | 40 | 1 | 40 | 1 | 40 | 1 | 40 |
| CTL | (0.79) **0.83** (0.87) | (0.96) **0.98** (0.99) | (0.79) **0.94** (1.13) | (0.14) **0.47** (1.61) | (0.23) **0.33** (0.45) | (0.04) **0.10** (0.16) | (0.16) **0.21** (0.27) | (0.001) **0.007** (0.015) |
| **CONMMPOL**_3KM | (0.77) **0.81** (0.85) | (0.96) **0.98** (0.99) | (0.76) **0.94** (1.16) | (0.12) **0.28** (0.65) | (0.13) **0.23** (0.33) | (0.03) **0.09** (0.14) | (0.18) **0.24** (0.30) | (0.001) **0.006** (0.011) |
| **CONMMPOL**_HR_12KM | (0.76) **0.80** (0.84) | (0.97) **0.98** (0.99) | (0.66) **0.82** (1.01) | (0.07) **0.14** (0.25) | (0.10) **0.20** (0.30) | (0.03) **0.006** (0.11) | (0.20) **0.25** (0.31) | (0.001) **0.003** (0.006) |
| **CONMMPOL**_12KM_3KM | (0.77) **0.81** (0.85) | (0.96) **0.98** (0.99) | (0.79) **0.96** (1.19) | (0.11) **0.26** (0.65) | (0.14) **0.23** (0.33) | (0.03) **0.08** (0.13) | (0.19) **0.25** (0.31) | (0.01) **0.005** (0.011) |


Table 8: Bootstrap 95% confidence intervals for verification statistics Forecast Accuracy (ACC), Frequency Bias (FBIAS),
Equitable Threat Score (ETS), False Alarm Ratio (FAR) and referred to experiments in line4. They are considered as a
function of thresholds (1mm/12h and 40mm/12h). The experiments are: CTL, CONMMPOLSPC_3KM,
CONMMPOLSPC_HR_12KM, CONMMPOLSPC_12KM_3KM.

| Experiment | ACC Thresholds mm/12h | | FBIAS Thresholds mm/12h | | ETS Thresholds mm/12h | | FAR Thresholds mm/12h | |
|---|---|---|---|---|---|---|---|---|
| | 1 | 40 | 1 | 40 | 1 | 40 | 1 | 40 |
| CTL | (0.79) **0.83** (0.87) | (0.96) **0.98** (0.99) | (0.79) **0.94** (1.13) | (0.14) **0.47** (1.61) | (0.23) **0.33** (0.45) | (0.04) **0.10** (0.16) | (0.16) **0.21** (0.27) | (0.001) **0.007** (0.015) |
| **CONMMPOLSPC**_3KM | (0.78) **0.82** (0.86) | (0.96) **0.98** (0.99) | (0.85) **1.03** (1.25) | (0.10) **0.27** (0.83) | (0.18) **0.28** (0.39) | (0.03) **0.07** (0.13) | (0.19) **0.25** (0.31) | (0.001) **0.005** (0.012) |
| **CONMMPOLSPC**_HR_12KM | (0.78) **0.82** (0.86) | (0.96) **0.98** (0.99) | (0.71) **0.86** (1.05) | (0.08) **0.22** (0.59) | (0.17) **0.28** (0.39) | (0.02) **0.06** (0.12) | (0.16) **0.21** (0.27) | (0.001) **0.005** (0.11) |
| **CONMMPOLSPC**_12KM_3KM | (0.77) **0.81** (0.85) | (0.96) **0.98** (0.99) | (0.87) **1.04** (1.28) | (0.09) **0.25** (0.70) | (0.16) **0.26** (0.36) | (0.04) **0.08** (0.14) | (0.22) **0.28** (0.34) | (0) **0.004** (0.009) |


Table 9: Bootstrap 95% confidence intervals for verification statistics Forecast Accuracy (ACC), Frequency Bias (FBIAS),
Equitable Threat Score (ETS), False Alarm Ratio (FAR) and referred to experiments in line 5. They are considered as a
function of thresholds (1mm/12h and 40mm/12h). The experiments are: CTL, CONMMPOLSPC3OL_3KM,
CONMMPOLSPC3OL_HR_12KM, CONMMPOLSPC3OL_12KM_3KM.

| Experiment | ACC Thresholds mm/12h | | FBIAS Thresholds mm/12h | | ETS Thresholds mm/12h | | FAR Thresholds mm/12h | |
|---|---|---|---|---|---|---|---|---|
| | 1 | 40 | 1 | 40 | 1 | 40 | 1 | 40 |
| CTL | (0.79) **0.83** (0.87) | (0.96) **0.98** (0.99) | (0.79) **0.94** (1.13) | (0.14) **0.47** (1.61) | **(0.23)** **0.33** (0.44) | **(0.04)** **0.10** (0.16) | (0.16) **0.21** (0.27) | (0.001) **0.007** (0.015) |
| **CONMMPOLSPC3OL**_3KM | (0.79) **0.83** (0.86) | (0.97) **0.98** (0.99) | (0.81) **0.96** (1.17) | (0.10) **0.24** (0.64) | (0.17) **0.27** (0.39) | (0.05) **0.12** (0.19) | (0.21) **0.27** (0.33) | (0) **0.003** (0.007) |
| **CONMMPOLSPC3OL**_HR_12KM | (0.78) **0.82** (0.86) | (0.96) **0.98** (0.99) | (0.77) **0.93** (1.13) | (0.13) **0.31** (0.86) | (0.20) **0.30** (0.41) | (0.004) **0.10** (0.17) | (0.14) **0.20** (0.26) | (0.002) **0.006** (0.012) |
| **CONMMPOLSPC3OL**_12KM_3KM | (0.79) **0.83** (0.86) | (0.97) **0.98** (0.99) | (0.82) **0.98** (1.18) | (0.08) **0.15** (0.24) | (0.19) **0.30** (0.41) | (0.04) **0.11** (0.18) | (0.19) **0.25** (0.31) | (0) **0.002** (0.003) |
