# Peer review of "Impact of Multiple Radar reflectivity data assimilation on the"

_Hydrology and Earth System Sciences, 2016_

## Referee Comment (RC1) · Anonymous Referee #1 · 18 Aug 2016

**General comments**

The manuscript describes data assimilation experiments of radar reflectivity observations with a view to improving the prediction of flash-flood events. This topic is an area of active research and it is at the core of hydro-meteorological research. Therefore, the paper undoubtedly fits within the scope of *HESS*.

However, the scientific significance of the paper seems quite low. The study is done for one single case, only one analysis is performed, and the tools (WRF ARW and its 3D-Var data assimilation system) are used without any clear novelty. The authors

claim that the novelty lies in the use of *multiple* radars in a *complex orography* to predict *intense precipitation*. Many studies have already addressed these points. WRF 3D-Var has been around for more than one decade and numerous reflectivity data assimilation studies of heavy-rainfall cases have been performed with it (e.g., Ha *et al.* 2011; Kiran Prasad *et al.* 2014; Maiello *et al.* 2014; Das *et al.* 2015, to cite a few of them), including with *multiple* radars and *complex orography* (e.g., Lee *et al.* 2010; Liu *et al.* 2013; Schwitalla and Wulfmeyer 2014). All the topics addressed in the manuscript have somehow been mentioned in previous studies. The impact of partial beam blockage has been reported by Chang *et al.* (2014). The sensitivity to the outer loops has been studied by Hsiao *et al.* (2012) and Maiello *et al.* (2014).

In the abstract, it is stated that the 'main goal is to establish a general methodology to quantitatively assess the performance of flash-flood numerical weather prediction at mesoscale'. However, I do not see anything in the manuscript but Observing System Experiments, which is a well-known framework to evaluate the impact of observations in a NWP system.

Furthermore, the results are limited to maps of quantitative precipitation forecasts and relating scores. Though statistical scores could suffice for a study bearing on a long time period, a single-case study should go more deeply into the data assimilation process and its relationship to the physics of the meteorological situation. For instance, what is the impact of the different configurations on the initial conditions? What is going on during the minimization step when the outer loop strategy is chosen? What are the consequences on the analysis? etc.

In short, I suggest the authors *i)* make their incremental contribution to the field more obvious and *ii)* investigate the data assimilation process itself directly and determine how the observations actually modify the initial state and what it implies for the short-range forecasts.

The organization of the paper could be improved. In section 2.2, assimilated observations are presented with some detail (amount of ingested observations), but the reader does not yet know in which domain(s) and at which hour(s) the data are assimilated. I suggest moving this section after the presentation of the model configurations (section 3.3).

**Specific comments**

- l 1: The title is misleading. The term 'Doppler' is used whilst no radial velocity is used. I suggest to drop it.

- l 16: I do not think that 'flash-flood numerical weather prediction' means anything sensible. In any case, there is no flash-flood forecasting in the manuscript.

- ll 31-34: The accuracy of model forecasts also depends on the model itself.

- l 53: Are there any references for 'HyMeX' and/or 'SOP1'?

- l 71: 'During IOP4' is not much specific. When did it occur?

- l 78: Are there any references for DEWETRA?

- ll 102-111: A table summarizing the characteristics of the radars would be welcome. Why is radial velocity not assimilated?

- ll 112-119: Is there any thinning applied to the reflectivity data? How is the conversion to the model format performed? Does it mean that the resolution of the data is the same as that of the model?

- l 130: The term 'independent' is at least ambiguous here because running the model over the innermost domain requires boundary conditions from the model

running over the outermost domain. So the run over the innermost domain actually depends on the run over the outermost domain.

- ll 150-152: Data assimilation is not restricted to 'atmosphere and ocean'.

- l 162: I do not understand the word 'fonts' in this context. Maybe 'sources' is meant here?

- l 165: What is 'pseudo' relative humidity?

- l 171: Can the authors briefly comment on the inconsistency between the drop size distributions in the observation operator and in the microphysics scheme of WRF?

- ll 200-205: The experiment names in the text and in table 2 are not consistent. What does LR (and later HR) mean?

- ll 221-241: If the goal of the paper is to evaluate the assimilation of radar data as the title suggests it, section 4.1 should shrink to one sentence or two at most, and figures 6 and 7 should be removed.

- ll 251-253: The authors should recall how the statistical indices are computed or mention references. I do not understand what is plotted in figure 9. When do the precipitation accumulations start and stop? It is written 'MEAN' and '2012-09-14 12:00:00 to 2012-09-16'. Does it mean that the scores are averaged over various forecast ranges?

- ll 251-253: The results are not as clear-cut as the authors claim it, and thus should be tempered. How many data points are used to compute the scores for 50 mm?

- ll 277-280: I do not understand why the ingestion of conventional observations produces the worst results.

- ll 307-309: Why is reflectivity in blocked regions neither corrected nor filtered out?

- ll 336-337: Well, the assimilation of radar data is already operational at several national meteorological services. The Korean Meteorological Administration has been doing it with WRF-3DVar since 2006! (see Xiao *et al.* 2008)

- ll 392-393: The paper has been published and the reference should be updated.

- l 397: 'Su' should be replaced with 'Sun'. The references should be checked carefully because there are other typos here and there.

- Fig 1: The quality (legibility) of the figure should be improved. Which model is shown? In the bottom panel, it should be specified what corresponds to isolines and colour shades, respectively.

- Fig 2: What do the coloured circles represent?

- Fig 3: Units and a scale are missing.

**References**

Chang, S.-F., J. Sun, Y.-C. Liou, S.-L. Tai, and C.-Y. Yang, 2014: The influence of erroneous background, beam-blocking and microphysical non-linearity on the application of a four-dimensional variational Doppler radar data assimilation system for quantitative precipitation forecasts. *Meteorological Applications*, **21**(2), 444–458. DOI: 10.1002/met.1439.

Das, M. K., M. A. M. Chowdhury, S. Das, S. K. Debsarma, and S. Karmakar, 2015: Assimilation of Doppler weather radar data and their impacts on the simulation of squall events during premonsoon season. *Natural Hazards*, **77**(2), 901–931. DOI: 10.1007/s11069-015-1634-9.

Ha, J.-H., H.-W. Kim, and D.-K. Lee, 2011: Observation and numerical simulations with radar and surface data assimilation for heavy rainfall over central Korea. *Advances in Atmospheric Sciences*, **28**(3), 573–590. DOI: 10.1007/s00376-010-0035-y.

Hsiao, L.-F., D.-S. Chen, Y.-H. Kuo, Y.-R. Guo, T.-C. Yeh, J.-S. Hong, C.-T. Fong, and C.-S. Lee, 2012: Application of WRF 3DVAR to operational typhoon prediction in Taiwan: impact of outer loop and partial cycling approaches. *Weather and Forecasting*, **27**(5), 1249–1263. DOI: 10.1175/waf-d-11-00131.1.

Kiran Prasad, S., U. C. Mohanty, A. Routray, K. K. Osuri, S. S. V. S. Ramakrishna, and D. Niyogi, 2014: Impact of Doppler weather radar data on thunderstorm simulation during storm pilot phase—2009. *Natural Hazards*, **74**(3), 1403–1427. DOI: 10.1007/s11069-014-1250-0.

Lee, J.-H., H.-H. Lee, Y. Choi, H.-W. Kim, and D.-K. Lee, 2010: Radar data assimilation for the simulation of mesoscale convective systems. *Advances in Atmospheric Sciences*, **27**(5), 1025–1042. DOI: 10.1007/s00376-010-9162-8.

Liu, J., M. Bray, and D. Han, 2013: A study on WRF radar data assimilation for hydrological rainfall prediction. *Hydrology and Earth System Sciences*, **17**(8), 3095–3110. DOI: 10.5194/hess-17-3095-2013.

Maiello, I., R. Ferretti, S. Gentile, M. Montopoli, E. Picciotti, F. S. Marzano, and C. Faccani, 2014: Impact of radar data assimilation for the simulation of a heavy rainfall case in central Italy using WRF–3DVAR. *Atmospheric Measurement Techniques*, **7**(9), 2919–2935. DOI: 10.5194/amt-7-2919-2014.

Schwitalla, T. and V. Wulfmeyer, 2014: Radar data assimilation experiments using the IPM WRF Rapid Update Cycle. *Meteorologische Zeitschrift*, **23**(1), 79–102. DOI: 10.1127/0941-2948/2014/0513.

Xiao, Q., E. Lim, D.-J. Won, J. Sun, W.-C. Lee, M.-S. Lee, W.-J. Lee, J.-Y. Cho, Y.-H. Kuo, D. M. Barker, D.-K. Lee, and H.-S. Lee, 2008: Doppler radar data assimilation in KMA's operational forecasting. *Bull. Amer. Meteor. Soc.* **89**(1), 39–43. DOI: 10.1175/BAMS-89-1-39.

---

## Referee Comment (RC2) · Anonymous Referee #2 · 30 Sep 2016

General Comments

The subject of the paper, describing experiments in radar data assimilation for a numerical weather prediction model to forecast a flash flooding event, is within the scope of HESS. This is an area where there is great potential for improved flood forecasting. I agree with many of the points made by the first anonymous referee. The area claimed as novelty in the paper, regarding the use of multiple radars with a high resolution forecast in complex orography, has been explored previously, as discussed by the first referee, and the study uses existing techniques and tools to analyse the case study presented. A stronger case for novelty could be made by exploring the meteorology of the case study in more depth, and making more explicit links to other work in

the HyMeX project. The conclusions are limited to the performance of different model configurations within the single case study, which may be useful in providing guidance for the development of a flood forecasting system in this context, but do not provide a significant contribution to the knowledge of the wider community. The methods used to investigate the case study are appropriate, but the case could have been explored in greater depth, with a greater focus on the interaction between the data assimilation system and the meteorology, for example an analysis of the data assimilation increments. Further, a number of statistical scores are produced, but the meaning of these indices, and their significance in a single case study, is not discussed. In a number of cases more detail could be given, for example the method by which radar data is mapped to the model grid. The paper does not indicate the use of Doppler velocity observations, and therefore the word Doppler should be removed from the title. The abstract is generally clear, although the stated aim to establish a general methodology to quantitatively assess the performance of a numerical weather prediction system exceeds what is demonstrated in the paper. I agree with the reviewer that the paper would be clearer if the model framework were introduced before section 2.2 on observations, which refers to the model domains before they have been introduced. There are a number of mistakes in the English spelling and structure of sentences, and I suggest that the paper would benefit from proof reading. I have identified some of these errors in the technical comments below. The literature review does not represent the breadth and extent of previous work in this area. The first reviewer has suggested a number of citations relating to previous work using WRF-3DVAR in particular. I would further suggest that the 3D-VAR technique used be put into the broader context of data assimilation methods used for radar data, such as latent heat nudging, 4D-VAR, and ensemble based methods, with consideration of the advantages and disadvantages of the particular technique used. A number of the figures are rather poor quality and could be improved by the use of higher resolution images. In summary, I would suggest:

(1) a detailed analysis of the meteorology of the case and the interaction of the data assimilation method with the model state,

(2) linking this paper more explicitly to its context within the HyMeX project,

(3) a more extensive literature review, considering the range of data assimilation methods which have been used for radar data,

(3) and proof reading the paper for quality of English.

Specific comments

Line 1: Remove Doppler from the title, this is misleading.

Line 20: "several damages" Be more specific, damage to buildings, infrastructure.

Lines 39:42 The citations here show an interest in the local context of the Adriatic region. If the focus of the paper is on building systems for flood forecasting in that region, for which this paper has greater potential to demonstrate novelty than in the demonstration of radar data assimilation techniques which already well documented in the literature, then this should be made more explicit, and the meteorology of the region and operational flood forecasting systems described in more detail.

Line 119: Please give some details of the format conversion.

Line 179: A comment on why 12 hour differences were used would be useful.

Lines 232:238 A number of statistical score values are listed, but the meaning and significance of these scores in these contexts is not considered.

Line 279: It is stated that a smaller number of conventional observations were ingested into the finer resolution domain than the coarser domain. Why is this? I would typically expect observations to be used at higher density with a finger resolution model.

Lines 306:310. I believe it is important to qualify the summary statement that the assimilation of conventional observations performed better with the coarser domain with the fact that more observations were used in the coarse domain model.

Line 336: I would describe the work in the paper as an interesting study in 3D-VAR

reflectivity assimilation in a flash flood case, which could be investigated further. I am not sure the paper presents a general approach. I agree that longer trials would be more informative, assuming that is what is meant by "pseudo-operational", as well as more in depth analysis of the meteorology and performance of the data assimilation system.

Figures 6, 8 and 10: These figures are quite low resolution and notably blurred. Sharper images would be clearer.

Technical corrections

Line 22: "multiple horizontal resolution" should be "multiple horizontal resolutions"

Line 22: "multiple radars data" should be "data from multiple radars"

Line 25: "rain gauges data" should be "rain gauge data"

Line 29: "recognized" recognizes

Line 30: "the very short term" "the" is unnecessary

Line 33: "subjected to" perhaps "limited by"?

Line 34: "Several researches" studies

Line 87: "It is worthwhile to point out" This is a redundant phrase and should be removed.

Line 93: intense precipitation which occurred

Line 94: "Zoom over CI target area" The zoom over the CI target area. Also CI is not defined.

Line 108: Define ISAC-CNR in the text

Line 112 "As is common knowledge" is rather informal and should be removed.

Line 176: "widely depends on the goodness" strongly depends on the quality?

Lines 185:186 This is unclear, I am not sure what is meant.

Line 214: "cover" coverage

Lines 248:250 "attempt" is not an appropriate word for describing a model forecast. Simply state whether a precipitation feature is forecast or not and evaluate its location and intensity.

Lines 264: 265 "Aiming to..." and "we start analyzing" is rather informal language.

―――――――――――――――――――――

---

## Author Comment (AC1) · 28 Oct 2016

**Responses to referee#1**

**General comments**

First of all we are grateful to anonymous referee for the great contribution to the manuscript coming from useful comments.

In this study we wanted to take advantage of the numerous Italian Intensive Observing Periods (IOPs) that affected the three Italian Target Areas (TAs) during the First Special Observation Period (SOP1) of the HyMeX campaign, but above all Central Italy (CI). Later, the choice fell on the IOP4 first of all because all the instruments activated was very successful (radar, sodar and microwave sensors were on alert in the Central Italy site from the evening Thursday 13 until Saturday 15 September 00UTC; extra operational soundings were performed on 13 September 18 UTC, 14 September 12 and 18 UTC in L'Aquila) and secondly it was a very interesting case with convective cells producing a remarkable amount of precipitation in a few hours (more than 150 mm) over Central Italy (Coastal Marche and Abruzzo) with precipitation peak of 300mm/24h. The event was quite well forecasted by all models operational during the campaign well in advance, but uncertainties remained until a few hours before the event regarding the exact location and amount of precipitation. On the other hand, we didn't find another Italian IOP, among those that have affected Central Italy, with so many radars activated simultaneously to enrich the analysis (for example during the IOP13 Monte Midia radar was out of service, whereas during the IOP16 Polar 55C was affected by some technical problems).

Concerning the novelty we claim in the paper, we know that many topics addressed in the manuscript have been already mentioned in previous studies, but except for Maiello et al. 2014, it is the first Italian experiment conducted on the Italian territory using the data of the Italian radars.

We are aware that constraining the analysis to maps of quantitative precipitation forecasts and relating scores could be a limit, but it was our choice to analyze the most important variable in a flash-flood event and to aim for the hearth of hydro-meteorological research. Nevertheless, we accept the advice to go deeply into the meteorology of the event to see which is its interaction with the data assimilation method.

We hope that the organization of the paper is now improved: section 2.2 has been moved after the presentation of the model configurations; section 4.1 has been shrink to few sentences and figures 6 and 7 have been removed; a table that summarizes the characteristics of the radars has been added. Moreover, several English mistakes have been corrected, the literature review has been updated and the quality of some figures has been improved. Also the title and the abstract have been modified.

**Specific comments**

**Line 1:** The word "Doppler" has been deleted and the title has been modified as follow: "Impact of Multiple Radar reflectivity data assimilation on the numerical simulation of a Flash Flood Event during the HyMeX campaign"

**Line 16:** The selected case study was tagged both as a Heavy Precipitation Event (HPE) and a Flash Flood Event (FFE). For this study we took advantage from all the instruments successfully activated during the event, with the aim of improving the forecast and alerting civil protection well in advance. In summary the objective here was to build a regionally-tuned numerical prediction model and decision-support system for civil prevention and protection within the central Italian regions. Moreover, the additional purpose is to find which type of observations (or a combination of several types) is more effective in improving the accuracy of the forecasted rainfall.

The sentence here has been modified as follows: "The main goal is to build a regionally-tuned numerical prediction model and decision-support system for civil prevention and protection within the central Italian regions, distinguishing which type of observations (or a combination of several types) is more effective in improving the accuracy of the forecasted rainfall."

**Lines 31-34:** We agree with the reviewer. The sentence has been modified as follows: " Nevertheless, the accuracy of the mesoscale NWP models is negatively affected by the "spin-up" effect (Daley 1991) and is mostly dependent on the errors in the initial and lateral boundary conditions (IC and BC), along with deficiencies in the numerical models themselves, and at the resolution of kilometers even more critical because of the lack of high resolution observations, beside for radar data."

**Line 53:** The references Ducrocq et al. 2014, Ferretti et al. 2014 and Davolio et al. 2015 have been added here.

**Line 71:** The sentence has been modified as follows " During the day of 14 September 2012 …. "

**Line 78:**  A reference for DEWETRA has been added both in the text and in the references list.

**Lines 102-111:**  A table that summerizes the characteristics of the three radars has been added and lines 102-111 have been rewrited as follows: " Volumetric reflectivity taken from three C-band Doppler radars operational during the IOP4 have been assimilated to improve IC. Radars have different technical characteristics and were operated with different scanning strategies and operational settings as shown in Table 1. Monte Midia (MM) and San Pietro Capofiume (SPC) radars are included in the Italian radar network, while Polar 55C (P55C) radar is a research radar working on demand which was operational during HyMeX IOPs (Roberto et al., 2016)."

We consciously decided to assimilate only reflectivity data, probably the term "Doppler" in the title was misleading (it has been dropped). A high quality of Doppler velocity is required for assimilation. However, quality of available data, especially due to the need of correct for aliasing was not suitable for assimilation in the case of the considered event. Therefore we have preferred assimilating only reflectivity.

**Lines 112-119:** Reflectivity data were quality controlled before ingested into the 3DVAR. However, an observation thinning before the minimization to avoid as much as possible error correlations between adjacent pixels is not performed because this procedure is not yet developed into WRFDA system for radar data. Nevertheless, a dynamical thinning has been devised that selects, for every assimilation cycle, the most influential partition of a particular measurement, from information based on the previous cycle: this is the multiple outer loops technique! (Cardinali et al. 2004, "Influence matrix diagnostic of a data assimilation system", Q. J. R. Meteorol. Soc., 130, 2827-2849). Indeed, the experiments performed using different numbers of outer loops allowed to compare the impact of a small sub-group of very influential data (i.e. radar observations, experiments with 3OL) on the forecast as the full amount of data. As future development, a thinning of radar data has to be undertaken either to reduce the observation-error spatial correlation or the computational cost of the assimilation (Montmerle and Faccani, 2009).

Concerning the data conversion to the model format, conventional and radar observations are treated in a different way. Conventional observational data are converted in LITTLE_R format using the Observation Preprocessor (OBSPROC) program provided by WRFDA system. The purposes of OBSPROC are to:

- Remove observations outside the specified temporal and spatial domains
- Re-order and merge duplicate (in time and location) data reports
- Retrieve pressure or height based on observed information using the hydrostatic assumption
- Check multi-level observations for vertical consistency and super adiabats
- Assign observational errors based on a pre-specified error file

- Write out the observation file to be used by WRFDA in ASCII or BUFR format

For what concern radar data, an ad hoc shell script in Fortran language has been written and adapted to each radar characteristics to perform conversion to the model format (more details about this have been added in the text).

**Line 130:** We agree with the reviewer. The sentence has been modified as follows: "a one-way nested configuration using *ndown* program is used"

**Lines 150-152:** We agree with the reviewer. The sentence has been modified as follows: " Data assimilation (DA), which applications arise in many fields of geosciences perhaps most importantly in weather forecasting and hydrology, in this context is the procedure by which observations are combined with the product (*first guess* or *background forecast*) of a NWP model and their corresponding error statistics to produce a bettered estimate (the *analysis*) of the true state of the atmosphere (Skamarock et al., 2008).

**Line 162:** The word "fonts" has been replaced by "sources".

**Line 165:** Pseudo relative humidity and total water mixing ratio are both control variables for the analysis of moisture observations in a global atmospheric data assimilation system. In a variational framework, the choice of control variable is important because the notion of "distance" between model and observations depends on it. A pseudo-relative humidity can be defined by scaling the mixing ratio by the background saturation mixing ratio. A pseudo-relative humidity analysis is shown to be equivalent to a mixing ratio analysis with flow-dependent variance specifications. The "pseudo" relative humidity is the water vapor mixing ratio divided by its saturated value in the background state.

**Line 171:** The microphysics scheme used is the New Thompson (Thompson et al., 2004). This scheme adopted a generalized gamma distribution shape for each hydrometeor species. The observational operator, on the other hand, uses the more simple Marshall and Palmer DSD which is an exponential one. This is a simplified gamma distribution, assuming 0 as exponent for the drop diameter. The main differences between the two DSDs are bounded where coalescence and evaporation processes and break-up process are active; these are the smallest and biggest drops region, i.e. the tails of the DSD. The difference introduced using these two DSDs plays a minor role respect to other errors like for example time and position shift.

**Lines 200-205:** The experiments names in the text and in table 2 are now consistent. The acronyms "LR" and "HR"mean low and high resolution respectively, in the sense that in the first case D01 is showed, D02 in the second case.

**Lines 221-241:** We agree with the reviewer. Section 4.1 has been rearranged as follows and figures 6 and 7 have been removed: "From the sensitivity test to different cumulus parameterization scheme (Table 2) the best performance is obtained by Grell3D scheme which is able to simulate the peak precipitation cumulated in 24 hours over Campo Imperatore, whereas KAIN-FRITSCH completely misses it (not shown here). The MET statistical analysis support the previous finding and the simulation  with *cugd_avedx* activated shows higher performances in terms of accuracy, equitable threat score and false alarm ratio than the other two simulations. Here after GRELL3D_MYJ_CUGD is referred as the control (CTL) experiment performed without any data assimilation. Therefore, a new set of simulations are performed following the previous strategies already mentioned in Section 4."

**Lines 251-253:** The statistical indexes have been calculated using the pointstat tool of MET (as reported in the lines 210-214). The MET Guide (Developmental Testbed Center, 2013: MET: Version 4.1 Model Evaluation Tools Users Guide. Available at http://www.dtcenter.org/met/users/docs/overview.php. 226 pp.) reports more details about the calculation of the statistical indexes. The reference will be added also in

lines 251-253. The Fig. 9 (now Fig.7) reports the statistical indexes for the 12 hours accumulated precipitation. The 12 hours accumulations have been calculated from the 2012-09-14 12:00:00 to 2012-09-16 00:00:00 every 6 hours, i.e. the scores of 2012-09-14 12:00:00 refers to the accumulated precipitation from 2012-09-14 00:00:00 to 2012-09-14 12:00:00. The word MEAN(9) on the title refers to the interpolation method used to match the gridded model output to the point observation. In details for this study the distance weighted mean in a 3 x 3 square has been used. The scores reported in Fig.9 have been averaged all over the data points belonging to the same threshold in the simulated time range.

**Lines 251-253:** The results will be tempered to make these clearer.
Fig. 1-2 report the histograms for the 12 hours accumulated precipitation. As you can see the first bin, including the precipitation lower than 10 mm/12h, is the most populated with approximately 20000 data points (Fig.1).

[Figure]

Fig 1. The histogram for the accumulated precipitation.

The Fig. 2 reports the same histogram removing the first bin to show how is crowded the following bins. The bin including the precipitations from 40 to 50 mm/12h has approximately 200 data points.

[Figure]

Fig 2. Zoom of the histogram for the accumulated precipitation (the first bin has been removed).

**Lines 277-280:** We found that when the assimilation is performed on the highest resolution domain only few SYNOP and even less TEMP fell down in the 3km domain at the analysis time of the assimilation procedure. For example after applying the WRFDA Observation Preprocessing procedure only a total of 338 observations (331 SYNOP and 7 TEMP) have been ingested into the D02 (Italy), compared to a total of 989 (967 SYNOP and 22 TEMP) into the D01 (Europe).

**Lines 307-309:** Since the three radars are managed by different organizations, a different radar data preprocessing procedure is followed and it depends on the case study.
Reflectivity is not corrected neither for total nor for partial beam blocking; nevertheless, all the data that are affected by partial beam blocking and clutter have been filtered out. In a future operational context, we could think to harmonize the processing of the three radars in order to achieve a spatially uniform quality.

**Lines 336-337:** We are aware that the assimilation of radar data is already operational at several meteorological services, but not in Italy. The Center of Excellence Cetemps (Abruzzo, Italy) is the only meteorological center in Italy that has radar data assimilation in operational mode since 2013, together with SYNOP and TEMP.

**Lines 392-393:** The reference has been updated.

**Line 397:** The reference has been corrected. Moreover, all the references have been checked both in the text and in the list.

**Fig.1:** The quality of figure 1 has been updated; a description of the meaning of isolines and colour shades has been added in the caption. The model used is WRF and the graphical tool GRADS.

**Fig.2:** The coloured circles represent the warning pluviometric thresholds  as follows:
[Figure]

The legend has been added in figure 2.

**Fig.3:** Figure 3 has been updated with units and scale.

---

## Author Comment (AC2) · 28 Oct 2016

**Responses to referee#2**

**General comments**

First of all we are grateful to anonymous referee for the great contribution to the manuscript coming from useful comments.

We wanted to take advantage of the numerous Italian Intensive Observing Periods (IOPs) that affected the three Italian Target Areas (TAs) during the First Special Observation Period (SOP1) of the HyMeX campaign, but above all Central Italy (CI). Later, the choice fell on the IOP4 first of all because all the instruments activated was very successful (radar, sodar and microwave sensors were on alert in the Central Italy site from the evening Thursday 13 until Saturday 15 September 00UTC; extra operational soundings were performed on 13 September 18 UTC, 14 September 12 and 18 UTC in L'Aquila) and secondly it was a very interesting case with convective cells producing a remarkable amount of precipitation in a few hours (more than 150 mm) over Central Italy (Coastal Marche and Abruzzo) with precipitation peak of 300mm/24h. The event was quite well forecasted by all models operational during the campaign well in advance, but uncertainties remained until a few hours before the event regarding the exact location and amount of precipitation. Moreover, we didn't find another Italian IOP, among those that have affected Central Italy, with so many radars activated simultaneously to enrich the analysis (for example during the IOP13 Monte Midia radar was out of service, whereas during the IOP16 Polar 55C was affected by some technical problems).

Concerning the novelty we claim, we know that many topics addressed in the manuscript have been already mentioned in previous studies, but except for Maiello et al. 2014, it is the first Italian experiment conducted on the Italian territory using the data of the Italian radars. Nevertheless, we accept the advice to go deeply into the meteorology of the event to see which is its interaction with the data assimilation method and making more explicit links to other work in the HyMeX project (i.e. Ducrocq et al. 2014, Davolio et al. 2015, Llasat et al. 2013).

We hope that the organization of the paper is now improved: section 2.2 has been moved after the presentation of the model configurations; section 4.1 has been shrink to few sentences and figures 6 and 7 have been removed; a table that summarizes the characteristics of the radars has been added. Moreover, several English mistakes have been corrected, the literature review has been updated and the quality of some figures has been improved. Also the title and the abstract have been modified.

**Specific comments**

**Line 1:** The word "Doppler" has been removed and the title has been modified as follow: "Impact of Multiple Radar reflectivity data assimilation on the numerical simulation of a Flash Flood Event during the HyMeX campaign"

**Line 20:** The sentence has been modified as follows: "causing several damages to buildings, infrastructures and roads".

**Lines 39-42:** We agree with the reviewer that the paper could have a great potential on demonstrate novelty if it is focused on building systems for flood forecasting in the central Adriatic region or central Italy in general. So the manuscript has been rearranged following this idea.

**Line 119:** Some details about radar format conversion has been added in the text as follows: "conversion to the model format is applied to all radars data (an ad hoc shell script in Fortran

language has been written and adapted to each radar characteristics)." See the response to a comment of referee1 for a detailed explanation about the format conversion of SYNOP and TEMP.

**Line 179:** The following sentence has been added in the text: "T+24 minus T+12 is typical for regional applications; it is important to include forecast differences to remove the diurnal cycle."

**Lines 232-238:** The statistical indexes used in this study are the ones commonly used for meteorological study, anyway you can find more details in the MET Guide (Developmental Testbed Center, 2013: MET: Version 4.1 Model Evaluation Tools Users Guide. Available at http://www.dtcenter.org/met/users/docs/overview.php. 226 pp.). The reference will be added in the lines 251-253.

**Line 279:** The meaning of the sentence here is the following: we found that when the assimilation is performed on the highest resolution domain only few SYNOP and even less TEMP fell down in the 3km domain at the analysis time of the assimilation procedure. For example after applying the WRFDA Observation Preprocessing procedure only a total of 338 observations (331 SYNOP and 7 TEMP) have been ingested into the D02 (Italy), compared to a total of 989 (967 SYNOP and 22 TEMP) into the D01 (Europe). In Italy (D02) we don't have a sufficiently dense observation network, above all of TEMP data.

**Lines 306-310:** We agree with the reviewer; the sentence has been modified as follows: "In summary, simulations results show that the assimilation of conventional data is better to perform on the lowest resolution domain because more observations were used in the coarser domain, whereas when the assimilation is performed on the highest resolution domain only few SYNOP and even less TEMP fell down in the 3km domain at the analysis time of the assimilation procedure."

**Line 336:** The sentence here has been rearranged as follows: "However, this work was an interesting study in 3D-Var reflectivity data assimilation that can encourage to investigate more flash flood cases occurred over central Italy, in order to make this proposed approach suitable to provide a realistic prediction of possible flash floods both for the timing and localization of such events. To confirm and consolidate these initial findings, apart from analyzing more case studies, a deeper analysis of the meteorology of the region and of the performance of the data assimilation system throughout longer trials in a "pseudo-operational" procedure is necessary."

**Figures 6, 8 and 10:** Figure 6 has been removed as suggested by referee1. Figures 8 and 10 have been improved.

**Technical corrections:**

**Line 22:** Done

**Line 22:** Done

**Line 25:** Done

**Line 29:** Done

**Line 30:** Done

**Line 33:** The sentence has been modified as follows: " the accuracy of the mesoscale NWP models is mostly dependent on "

**Line 34:** Done

**Line 87:** Done

**Line 93:** Done

**Line 94:** Done. The acronym CI has been already defined in line 68.

**Line 108:** Done

**Line 112:** Done

**Line 176:** The sentence has been modified as follows: " strongly depends on the quality "

**Lines 185-186:** The sentence has been modified as follows: " The previous coarser resolution WRF forecast at 00:00UTC is used as the first guess (FG) in the 3D-Var experiment, because 00:00UTC has been selected as the "*analysis time*" of the assimilation procedure."

**Line 214:** Done

**Lines 248-250:** The sentence has been modified as follows: "Observing the outputs of different experiments (Fig. 8) listed in Table 2, best simulation is found for CONMMPOLSPC_LR_12KM (black arrow in Fig.8e): the rainfall maximum over Campo Imperatore is very well simulated, however a cell displacement is noticeable. Furthermore the precipitation feature along the coasts (black oval) is also forecasted."

**Lines 264-265:** The sentence has been modified as follows: "In order to investigate the impact of the assimilation at different resolutions, we analyzes.... "

---

## Author Comment (AC3) · 10 Nov 2016

**Lines 336-337:** We are aware that the assimilation of radar data is already operational at several meteorological services, but the Center of Excellence Cetemps is one of the few meteorological centers in Italy that has radar data (volumetric reflectivity and radial velocity) assimilation in operational mode, together with SYNOP and TEMP observations, using the 3D-Var assimilation technique. Also the Italian ARPA-SIMC operationally performs the assimilation of radar-derived precipitation rates using the latent heat nudging into the COSMO model and, as future step in the next year, the technique will be extended to the direct use of 3-D radar data (radial wind and reflectivity).

---

## Referee Report (RR1)

**Review of 'Impact of Multiple Radar reflectivity data assimilation on the numerical simulation of a Flash Flood Event during the HyMeX campaign' by Maiello *et al.**

**General comments**

The authors have improved their manuscript by clarifying their goal and their contribution to the field of hydro-meteorological research.

I am pleased to read that the authors 'accept the advice to go deeply into the meteorology of the event to see which is its interaction with the data assimilation method'. However, I do not see much evidence of it in the revised manuscript.

My opinion is that without any clear statistical significance (see below my comment regarding confidence intervals) or in-depth analysis of the data assimilation process, the manuscript fails to meet publication standards.

**Specific comments**

Most of my previous specific comments have been addressed satisfyingly. I list here below those that still need to be addressed.

- Subsection 3.1: In my previous review, I asked for more details regarding the assimilated radar observations. I still do not understand what exactly is being done. The authors replied that no thinning was performed. I think that this piece of information should be mentioned in the text.

  I do not know what 'model format' means (l 171). Does it mean that the radar data are interpolated onto the model grid? If yes, how? Is there any smoothing? What is the minimum assimilated reflectivity? Does it depend on the range?

  It should be added in the text that pixels affected by partial beam blockage have been removed, as mentioned by the authors in their reply to one of my comments.

- ll 219-222: The reader wonders which experiment is actually selected. I suggest moving the contents of Subsection 4.1 right after ll 219-222 (and remove the subsectioning of Section 4 or rename the current Subsection 4.2 as a new Section 5). So that MET is already introduced, ll 239-245 could form the contents of a Subsection 3.3 titled, eg, 'Evaluation'.

- ll 264-266: The details given by the authors regarding how the statistical indices are computed ('The 12 hours accumulations have been calculated from the 2012-09-14 12:00:00 to 2012-09-16 00:00 every 6 hours') should be added to the text.

It seems that MET also provides bootstrap confidence intervals. It would be useful to consider them when discussing the results.

- ll 292-293: I still do not understand why CON_3KM is worse than CTL (it seems quite obvious from Table 5 or Table 7). The authors explain that there are only few data ingested in the smaller domain. But it is anyway more than no data as in CTL, isn't it? Also, why does data assimilation in both domains (experiment CON_12KM_3KM) produce low statistics compared to no assimilation at all (CTL) or assimilation in the coarser-resolution domain only (CON_HR_12KM)?

- Fig 1: The source of the data (most likely analyses of a global model, I suppose) should be mentioned.

- Table 2: Points ('.') should be used instead of commas (',') as decimal separators. The SI symbol for kilometre is 'km', not 'Km'. Degree symbols ('°') should be added after elevation angles (I suppose degrees are actually used here).

---

## Referee Report (RR2)

**Review of 'Impact of Multiple Radar reflectivity data assimilation on the numerical simulation of a Flash Flood Event during the HyMeX campaign' by Maiello *et al.**

**General comments**

I do acknowledge the efforts made by the authors to address my comments. However, some new elements have been introduced in the manuscript that call for clarifications:

- In my previous review, I suggested using confidence intervals to evaluate the statistical significance of the results. The authors have computed these confidence intervals, which add value to the manuscript. However, the results are presented in a cumbersome way. The authors provide 9 (!) tables with skill scores and associated confidence intervals. The authors should pick up salient features in these tables and explicitly refer to them in the text to help the reader (and convince him/her!; see some related comments here below, *which are not exhaustive*). In particular, the most striking (and not much discussed) feature is that nearly all figures have overlapping confidence intervals, which definitely call for cautious interpretations and justify backing up any conclusion carefully.

- The use of a dynamical thinning in relationship with the outer loop technique needs to be clarified (see my comment below). In the end, are there more or less radar data ingested with this technique?

**Specific comments**

- Section 2.1: It should be mentioned in the text that Figure 2 was produced with DEWETRA. Otherwise, the reader who overlooks Figure 2's caption does not understand why DEWETRA is introduced here.

- Section 3.1: The authors explain that 'volume reflectivity radar data, for each elevation, are projected onto the Cartesian plane in order to find the closest radar bin for each Cartesian grid point and then they are interpolated by the 3D-Var code of WRF'. This is still unclear to me. Does it mean that there is a radar observation assimilated at every model grid point (that of 'the closest radar bin')? What kind of interpolation is done by the 3D-Var code? In other words, the interesting (and missing) piece of information here is the *spatial resolution* of the observations.

- Section 3.1: The authors write: 'Moreover, no observation thinning is performed because this procedure is not yet developed into the 3D-Var system

for radar data. Nevertheless, a dynamical thinning has been devised that selects, for every assimilation cycle, the most influential partition of a particular measurement, from information based on the previous cycle: this is the multiple outer loops technique explained later in Section 4.' I have a different understanding of the outer loop technique. I understand that it is meant to update linearised operators (such as the observation operator) during the minimization process. As a consequence, more observations are assimilated with each iteration and the quality of the analysis is improved.

I do not see the relationship between the outer loop technique and thinning. The purpose of the latter is to counterbalance the use of an overly simplistic (ie, diagonal) observation error covariance matrix or to reduce the computational cost of the assimilation. Thinning actually results in reducing the amount of observations.

So what is the 'dynamical thinning [that] has been devised that selects, for every assimilation cycle, the most influential partition of a particular measurement, from information based on the previous cycle'? In the cited literature, Rizvi et al. (2008) pertains to the outer loop technique (in passing, it may be more appropriate to cite peer-reviewed articles such as Hsiao et al. 2012), and Liu and Rabier (2002) pertains to thinning, but no citation refers to both thinning and outer loop.

The sentence in question is almost a verbatim excerpt from Cardinali (2013, 2014)[1]. Does it mean that the authors used a dynamical thinning based on the influence matrix, which is the topic dealt with by Cardinali (2013, 2014)? In that case, they should add a reference to the technique they used or give more details about how it works. If this is related to the outer loop technique, the authors should formulate this relationship more explicitly.

- Section 5, comments on Table 4: Table 4 contains a lot of figures and the conclusions which are drawn from it are that the values are 'good' for ACC and FAR (which is expected when the considered events are rare) and that the experiments overestimate light precipitation. Is Table 4 really needed?

- Section 5: What message do the authors want to convey with the following sentence: 'MET indices in Table 5 suggest that CTL and CON_HR_12KM
* * *
[1]Compare (common terms are highlighted in bold face):

> Nevertheless, **a dynamical thinning** has been devised **that selects,** for **every assimilation cycle, the most influential partition of a particular measurement, from information based on the previous cycle**: this is the multiple outer loops technique explained later in Section 4. (Maiello et al. 2016)

and:

> In this case, **a dynamical thinning** can be thought/considered **that selects,** at **every assimilation cycle, the most influential measurement partition of a particular** remote sensing instrument, **from information based on the previous cycle** (see also Rabier et al., 2002). (Cardinali 2013, 2014, p 158)

have the widest spread between the CIs limits for higher thresholds'?

- Section 5, l 327: I do not understand how the conclusion that 'CON-MMPOLSPC3OL_HR_12KM is the simulation with the best response' is reached. The score values for all experiments are quite close to each other and within the uncertainty intervals, and CONMMPOLSPC3OL_HR_12KM even scores lower than CTL for ACC(1 mm), FBIAS and ETS(1 mm).

- Section 5, ll 338-339: I do not understand that 'the frequency of rainfall overestimation for higher thresholds has been reduced by radar reflectivity assimilation performed only on D01'. For higher thresholds, FBIAS is systematically below 1, which means that the experiments *under*estimate the frequencies of large rainfall accumulations. The underestimations are even worse when radar reflectivity data are assimilated in D01 only: all FBIAS score values lie below .31 when radar reflectivity is assimilated, vs .47 and .49 for CTL and CON_HR_12KM, respectively.

- Section 5, ll 342-344: 'The assimilation, operated on both 12 km and 3 km, gives better results than the ones on column 1, but a worse response than the others on column 2 is given for higher thresholds.' Could the authors please back this up? It is far from straightforward to see it.

- Section 5, ll 378-380: How can shielded radar data lead to underestimating precipitation forecasts? I understood that they had been filtered out (see ll 184-185 'all the data that are affected by partial beam blocking and clutter have been filtered out').

**References**

Cardinali, Claudia (2013). *Observation influence diagnostic of a data assimilation system*. URL: http://www.ecmwf.int/sites/default/files/elibrary/2013/16938-observation-influence-diagnostic-data-assimilation-system.pdf.

— (2014). 'Observation influence diagnostic of a data assimilation system'. In: *Advanced Data Assimilation for Geosciences: Lecture Notes of the Les Houches School of Physics: Special Issue, June 2012*. Ed. by É. Blayo et al. Lecture Notes of the Les Houches Summer School. Oxford University Press. Chap. 5, pp. 137–163. ISBN: 9780198723844. URL: https://books.google.it/books?id=RSXSBAAAQBAJ.

Hsiao, Ling-Feng et al. (2012). 'Application of WRF 3DVAR to Operational Typhoon Prediction in Taiwan: Impact of Outer Loop and Partial Cycling Approaches'. In: *Wea. Forecasting* 27.5, pp. 1249–1263. ISSN: 1520-0434. DOI: 10.1175/waf-d-11-00131.1. URL: http://dx.doi.org/10.1175/WAF-D-11-00131.1.

Liu, Z.-Q. and F. Rabier (2002). 'The interaction between model resolution, observation resolution and observation density in data assimilation: A one-dimensional study'. In: *Quart. J. Roy. Meteor. Soc.* 128.582, pp. 1367–1386. DOI: 10.1256/003590002320373337.

Maiello, Ida et al. (2016). 'Effects of Multiple Doppler Radar data assimilation on the numerical simulation of a Flash Flood Event during the HyMeX campaign'. In: *Hydrology and Earth System Sciences Discussions*, pp. 1–25. DOI: 10.5194/hess-2016-320.

Rizvi, Syed R. H. et al. (2008). 'Impact of outer loop for WRF data assimilation system (WRFDA)'. In: *9th WRF Users' Workshop.* (Boulder, CO, 23rd–27th June 2008).

---

## Referee Report (RR3)

**Review of 'Impact of Multiple Radar reflectivity data assimilation on the numerical simulation of a Flash Flood Event during the HyMeX campaign' by Maiello *et al.**

**General comments**

The manuscript has been substantially improved and I recommend it to be published once the remaining minor comments here below are addressed.

**Specific comments**

- Is 'a non-Gaussian error probability density function' (ll 74-75) the reason why the 'simulation' (l 73) of radar reflectivity is challenging? The authors maybe meant 'assimilation' instead of 'simulation'.

- l 194: Is 'altitude' meant here instead of 'elevation'? I suppose the WRF-3D-Var system does not know about the radar elevation angles, position, etc.

- ll 288-292: The answers of the authors to my previous review partly clarified how the radar data are processed in the assimilation system.

  I understand that all radar observations enter the WRF-3D-Var system. Some of them are rejected, I suppose based on observation-minus-guess departures. The outer loop technique allows to increase the amount of assimilated data at each iteration.

  Could the authors give an example or order of magnitude of: i) the amount of radar data that enter the 3D-Var system, ii) the fraction of radar data that are rejected, say, at the first iteration and at the last one, respectively? That would show the efficiency of the outer loop technique in assimilating more data and give keys to understand the differences between experiments using this technique or not.

- Seven tables of verification statistics (out of nine in the previous version) are still present in this new version. I leave it to the editor to decide whether it is acceptable or should either be reduced or converted into figures for improved legibility.

---

## Author Response (AR2)

**Review of 'Impact of Multiple Radar reflectivity data assimilation on the numerical simulation of a Flash Flood Event during the HyMeX campaign' by Maiello et al.**

**Responses to referee#2**

**General comments**

The authors have improved their manuscript by clarifying their goal and their contribution to the field of hydro-meteorological research. I am pleased to read that the authors 'accept the advice to go deeply into the meteorology of the event to see which is its interaction with the data assimilation method'. However, I do not see much evidence of it in the revised manuscript. My opinion is that without any clear statistical significance (see below my comment regarding confidence intervals) or in-depth analysis of the data assimilation process, the manuscript fails to meet publication standards.

We thank again the reviewer for the useful comments. The authors worked on the manuscript to provide a clearer statistical significance (the results have been interpreted in the light of bootstrap confidence intervals) and trying to go more in-depth into the assimilation process. Please refer to the answers below and to the modifications in the manuscript highlighted in green.

Moreover, some other appropriate references have been included in the text and also the presentation quality has been improved (some sections have been rearranged in a well structured way and English language has been corrected).

**Specific comments**

**Subsection 3.1:** In my previous review, I asked for more details regarding the assimilated radar observations. I still do not understand what exactly is being done. The authors replied that no thinning was performed. I think that this piece of information should be mentioned in the text. I do not know what 'model format' means (l 171). Does it mean that the radar data are interpolated onto the model grid? If yes, how? Is there any smoothing? What is the minimum assimilated reflectivity? Does it depend on the range? It should be added in the text that pixels affected by partial beam blockage have been removed, as mentioned by the authors in their reply to one of my comments.

Subsection 3.1 has been rearranged mentioning in the text more details about the assimilated radar observations, the piece of information about thinning has been added to the text, the sentence l171 has been better explained and also the information concerning the partial beam blockage has been included into the manuscript.

Following we tried to explain the meaning of the sentence in line 171:

Volume reflectivity radar data, for each elevation, are converted onto the Cartesian plane in order to find the closest radar bin for each Cartesian grid point. Then, they are interpolated by the 3D-Var code of WRF.

No smoothing or superobbing is applied.

The minimum assimilated reflectivity is related to the minimum detectable reflectivity (MDZ) that depends on the distance as well as the technical parameters of each radar system. Anyway a minimum threshold is set to -20 dBZ.

In addition, due to the fact that non conventional data such as radar data (or for instance also radiance and rainfall data) don't go through the OBSPROC procedure, they require separate pre-processing. Because radar data comes in a variety of different formats, it is the user's responsibility to convert their data into this format. For 3D-Var, these observations should be placed in a file named *ob.radar*. So it was necessary to write an ad hoc shell script in Fortran language to convert the native radar format into the proper format (a text-based format) for the ingestion into the 3D-Var. This format is showed following:

Subsection 3.1 has been modified as follows:

Conventional observations SYNOP and TEMP were retrieved from the ECMWF Meteorological Archival and Retrieval System (MARS). They have been packed in a suitable format for ingest into the assimilation procedure using the Observation Preprocessor (OBSPROC) module provided by the 3D-Var system. Among its main functions there are also to perform a quality control check and to assign observational errors based on a pre-specified error file. In short, a total of 983 observations (967 SYNOP and 16 TEMP) are ingested into the coarse resolution domain, whereas a total of 338 (333 SYNOP and 5 TEMP) observations into the high resolution one.
Reflectivity volumes taken from three C-band Doppler radars operational during the IOP4 have been assimilated to improve IC. The radars have different technical characteristics and were operated with different scanning strategies and operational settings as shown in Table 1.
Monte Midia (MM) and San Pietro Capofiume (SPC) radars are included in the Italian weather radar network, while Polar 55C (P55C) radar is a research radar working on demand, butwas operational during the IOPs of the HyMeX campaign (Roberto et al., 2016).
It is worth mentioning that radar data can be affected by numerous sources of errors, mainly due to ground clutter, attenuation due to propagation or beam blocking, anomalous propagation and radio interferences. This is the reason why a preliminary "cleaning" procedure is applied to the measured radar reflectivity from the three radars before the assimilation process, consisting of the following 3 steps:
- a first quality check of radar volumes to filter out radar pixels affected by ground clutter and anomalous propagation. Furthermore, Z was corrected for attenuation using a methodology based on the specific differential phase shift ($K_{dp}$) available for dual polarization radars (Vulpiani et al, 2015); moreover, reflectivity is not corrected for partial beam blocking : all the data that are affected by partial beam blocking and clutter have been filtered out;

- **ll 219-222:** The reader wonders which experiment is actually selected. I suggest moving the contents of Subsection 4.1 right after ll 219-222 (and remove the subsectioning of Section 4 or rename the current Subsection 4.2 as a new Section 5). So that MET is already introduced, ll 239-245 could form the contents of a Subsection 3.3 titled, eg, 'Evaluation'.

We agree with the reviewer, the modifications suggested can improve the reading of the manuscript, therefore the content of subsection 4.1 have been moved after ll 219-222 and subsection 4.2 has been renamed as a new Section 5. Moreover, ll 239-245 formed the contents of a new Subsection 3.3 titled "Evaluation" where an overview of MET tool and bootstrap confidence intervals method have been done.

- **ll 264-266:** The details given by the authors regarding how the statistical indices are computed ('The 12 hours accumulations have been calculated from the 2012-09-14 12:00:00 to 2012-09-16 00:00 every 6 hours') should be added to the text.

The details regarding how the statistical indices are computed have been added to the text as follows (the authors decided to substitute the old figure 7 with a new table 4 that summarizes the realized scores):

It seems that MET also provides bootstrap confidence intervals. It would be useful to consider them when discussing the results.

For the calculation of MET statistical indices the bootstrap confidence intervals (CIs 95%) have been used.

The values of any statistic are realizations from a population of possible values and single summary scores gives only an indication of the forecast performance, whereas the CIs give us information about how much aleatoric (or sampling) uncertainty we have. Providing and discussing CIs can strengthen the results in the sense that they provide uncertainty information, which gives us a better idea about whether or not, in this case, the values obtained are likely good or bad.

- **ll 292-293:** I still do not understand why CON 3KM is worse than CTL (it seems quite obvious from Table 5 or Table 7). The authors explain that there are only few data ingested in the smaller domain. But it is anyway more than no data as in CTL, isn't it? Also, why does data assimilation in both domains (experiment CON 12KM 3KM) produce low statistics compared to no assimilation at all (CTL) or assimilation in the coarser resolution domain only (CON HR 12KM)?

The doubts of the reviewer would be reasonable, although in these lines we were focusing the attention on the impact of the assimilation at different resolution and not on the impact of different types of observations.

Anyway, it is evident from the images below how only one of the 5 total TEMP (yellow stars) falls into the 3 km domain (figure B) and also how far it is from where the event occurred: this is the sounding of Bologna in Emilia Romagna region. Furthermore, looking at Abruzzo and Marche regions, or central Italy in general, there is a low density of surface stations. It has to be reminded that most of these observations have already been used by ECMWF to produce their analysis and that they are here used as first guess, even if at lower resolution (0.25°): therefore, they come to be correlated to the background and the improvements of those experiments where they are assimilated are expected to be low. For this reason we couldn't expect a large improvement in the DA experiments where both SYNOP and TEMP have been assimilated. As explained by Liu and Rabier (2002) in their paper entitled "The interaction between model resolution, observations resolution and observations density in data assimilation: a one-dimensional study", the correct balance among the model resolution, the observation density and the observation resolution is crucial for a good initialization. In their paper Liu and Rabier used a synthetic set of data, varying the density and resolution in a one dimensional case: they found that for these observations having a spatial error correlation, the thinning process could help to find a good compromise between the data density and correlation, producing a good analysis accuracy. In our work the bi-dimensional data density (SYNOP) is well larger than that of the three-dimensional ones (TEMP). So the poor results obtained assimilating conventional observations probably depend on the large difference between the spatial density and the number of surface bi-dimensional and three-dimensional data of radiosondes.

The above considerations have been also added to the manuscript at the end of new section 5.

[Figure]

[Figure]

[Figure]

**967 SYNOP  (dots) + 16 TEMP (stars) on D01 - 00UTC 14 SEPT 2012**      **331 SYNOP  (dots) + 5 TEMP (stars) on D02- 00UTC 14 SEPT 2012**

- **Fig.1:** The source of the data (most likely analyses of a global model, I suppose) should be mentioned.

  We agree with the reviewer. The caption of figure 1 has been modified as follows:

  "Figure 1. ECMWF (European Center for Medium-Range Weather Forecasts) analyses at 12:00UTC on 14 September 2012: a) mean sea level pressure, c) temperature (color shades) and geopotential height (black isolines) at 500 hPa; ECMWF analyses at 12:00UTC on 15 September 2012: b) mean sea level pressure, d) temperature (black isolines) and geopotential height (color shades) at 500 hPa."

- **Table 2:** Points ('.') should be used instead of commas (',') as decimal separators. The SI symbol for kilometre is 'km', not 'Km'. Degree symbols ('°') should be added after elevation angles (I suppose degrees are actually used here).

  I suppose the reviewer means Table 1 instead of Table 2. Commas (',') as decimal separators have been replaced by points ('.'); the SI symbol for kilometre became "km"; degree symbols ('°') have been added after elevation angles.

[revised manuscript text omitted]

- the minimum assimilated reflectivity is set to -20 dBZ;

After the pre-processing procedure, a conversion from the native radar format into the one requested for the ingestion into the 3D-Var is applied to all radars reflectivity data.

Moreover, no observation thinning is performed because this procedure is not yet developed into the 3D-Var system for radar data. Nevertheless, a dynamical thinning has been devised that selects, for every assimilation cycle, the most influential partition of a particular measurement, from information based on the previous cycle: 
[revised manuscript text omitted]

In Table 4 statistical indices ACC, FBIAS, ETS and FAR are reported, with their relative upper and lower confidence limits for the 12 hours accumulated precipitation and for two thresholds of precipitation, namely 1 mm and 40 mm, for light and heavy rain regimes, respectively. These two thresholds have been chosen due to their higher statistical significance than the other ones.

We obtained likely good values for ACC and FAR for all the experiments and for heavy rain regimes, strengthened by a small uncertainty interval. On the other hand, for the lower threshold it can be seen that for all simulations the values of FBIAS considering also the confidence intervals are greater than one. One possible interpretation of the impact of the

lower threshold, is that with 95% confidence all the experiments are overestimating the frequency of precipitation around 1 mm/12h.

Similarly to the above comparison, presented in figure 7 are high resolution results (HR) obtained performing reflectivity assimilation on 12 km domain (column 1), on 3 km (column 2) and on 12 km and 3 km together (column 3); to the top of figure 7 the CTL experiment on D02 is shown. Figure 7 is organized as follows: viewing panels by line, on line 1 all the simulations with conventional data assimilation only (CON*) are found; on line 2 all the experiments with the assimilation of the reflectivity data from MM radar added (CONMM*); on line 3 all the experiments with the assimilation of the reflectivity data from 2 C-band radars added (CONMMPOL*); on line 4 all the experiments with the assimilation of the reflectivity data from all 3 C-band radars added (CONMMPOLSPC*); on line 5 the simulations where the strategy of outer loop is adopted (CONMMPOLSPC3OL*). In order to quantify the uncertainty associated to these experiments, the bootstrap 95% confidence intervals for verification statistics ACC, FBIAS, ETS, FAR have been summarized over tables (from 5 to 12) reporting again the two thresholds of precipitation: 1 mm/12h and 40 mm/12h (light and heavy rain regimes respectively).

In order to investigate the impact of the assimilation at different resolutions, we analyze figure 7 by column and comparing it with the available observations (Fig. 2) using also the statistical analysis:

- column 1 (12KM): CTL produces an overestimation of the rainfall that is not corrected by the assimilation of conventional data, but assimilating the reflectivity from the 3 radars and introducing the 3 outer loops (Fig. 7 column 1 line 5) the main cells are better reproduced. MET indices in Table 5 suggest that CTL and CON_HR_12KM have the widest spread between the CIs limits for higher thresholds, whereas CONMMPOLSPC3OL_HR_12KM is the simulation with the best response, secondly CONMM_HR_12KM, if we consider both the estimate of the scores and their uncertainty;
- column 2 (3KM): a partial correction of the rainfall overestimation compared to column 1 is observed especially if reflectivity from all the radars are assimilated and the outer loop strategy is applied; the statistical indices in Table 6 show CONMMPOLSPC3OL_3KM as the best experiment among the assimilated ones because of competitive values of ACC at both thresholds and FBIAS and FAR for the light and heavy rain thresholds, respectively;
- column 3 (12KM_3KM): rainfall overestimation was partially corrected compared to columns 1 and 2 by all the experiments; the MET statistics in Table 7 shows that CTL and CONMMPOLSPC3OL_12KM_3KM are the experiments with better values and small uncertainty, especially for ACC and ETS scores, although there is a quite broad spread in FBIAS of CTL experiment if we consider higher thresholds.

Summarizing, the previous analysis suggests that the frequency of rainfall overestimation for higher thresholds has been reduced by radar reflectivity assimilation performed only on D01. Furthermore, improvements come out for heavy rain regimes when radar reflectivity assimilation has been performed on the highest resolution domain, whereas the ingestion of conventional observations produces the worst results since a smaller number of them were assimilated into the finest resolution domain (for instance one sounding on five total) than that the coarser one. The assimilation, operated on both 12 km and 3 km, gives better results than the ones on column 1, but a worse response than the others on column 2 is given for higher thresholds.

In order to examine the impact of the assimilation of different data and radars, we can now analyze the experiments showed in figure 7 line by line. The results are compared with the observations of Fig. 2. The following considerations are worth discussing:

- line 1 (CON): a strong reduction of the rainfall is found with respect to CTL if conventional data are assimilated, but the rainfall pattern remains unchanged. Statistical indices of CON experiment (Table 8) do not improve the performances of CTL (despite a reduction in some cases of the spread between the CIs limits for higher thresholds of the FBIAS). The indices values suggest a slightly better performance when the conventional observations are assimilated only on the bigger domain and for higher thresholds, together with an improvement of FAR index for heavy rain regime;

- line 2 (CONMM): a further reduction in the precipitation overestimation is found as well as some variations in the pattern of the rainfall; the scores in Table 9, together with their bootstrap upper and lower limits, show that MM radar reflectivity and conventional observations assimilation, improves the model performance above all for lower thresholds respect to the experiments where only SYNOP and TEMP were ingested. It applies also for some of the scores at higher thresholds;

- line 3 (CONMMPOL): a quite strong improvement in the rainfall amount is found for all simulations. However, from the statistics of Table 10 we found a general worsening of the results both for light and heavy rain regimes when POL is added (ACC, FBIAS and ETS);

- line 4 (CONMMPOLSPC): a clear correction of the rainfall pattern is found; the overestimation produced by the simulation where the reflectivity from all the radars are assimilated on the 3 km domain has been corrected by the experiment in which the reflectivity is assimilated both on D01 and D02; the uncertainty in the realized scores of Table 11 suggests that the addition of SPC radar improves the results, furthermore they are not better than those where only MM is ingested;

- line 5 (CONMMPOLSPC3OL): the outer loop experiment confirms the strong overestimation reduction by *12KM_3KM; from Table 12 it seems that the introduction of 3OL improves the indices estimate and bounds above all when the 12 km domain is considered; CONMMPOLSPC3OL_12KM_3KM can be seen as the best simulation taking into account all the verification scores at both rainfall thresholds.

In summary, simulations results show that assimilation of conventional data is better to perform on the lowest resolution domain because more observations were used in the coarser domain, whereas when the assimilation is performed on the highest resolution domain only few SYNOP and even less TEMP fell down in the 3 km domain at the analysis time of the assimilation procedure. The impact of the conventional observations are expected to be lower than those of the non conventional ones, because most of them have already been used by ECMWF to produce their analysis and that they are here used as first guess, even if at lower resolution (0.25°). Therefore, they result to be correlated to the background and the improvements of those experiments where they are assimilated are expected to be low.

With regard to the assimilation of reflectivity radar data, should be noted that P55C radar observation is shielded at the lowest elevations by the Apennines. This leads to an underestimation of the precipitation, especially when the peak occurs; as a consequence a wrong estimation is given to the WRF model worsening the 
[revised manuscript text omitted]

**0.83**
(0.86) | (0.97)
**0.98**
(0.99) | (0.82)
**0.98**
(1.18) | (0.08)
**0.15**
(0.24) | (0.19)
**0.30**
(0.41) | (0.04)
**0.11**
(0.18) | (0.19)
**0.25**
(0.31) | (0)
**0.002**
(0.003) |

---

## Author Response (AR3)

**Review of 'Impact of Multiple Radar reflectivity data assimilation on the numerical simulation of a Flash Flood Event during the HyMeX campaign' by Maiello et al.**

**General comments**

I do acknowledge the efforts made by the authors to address my comments. However, some new elements have been introduced in the manuscript that call for clarifications:

In my previous review, I suggested using confidence intervals to evaluate the statistical significance
of the results. The authors have computed these confidence intervals, which add value to the
manuscript. However, the results are presented in a cumbersome way. The authors provide 9 (!)
tables with skill scores and associated confidence intervals. The authors should pick up salient
features in these tables and explicitly refer to them in the text to help the reader (and convince
him/her!; see some related comments here below, which are not exhaustive). In particular, the
most striking (and not much discussed) feature is that nearly all figures have overlapping
confidence intervals, which definitely call for cautious interpretations and justify backing up any
conclusion carefully.

We agree with the reviewer that 9 tables, with 4 different skill scores and 2 thresholds of precipitation and related confidence intervals, could make the paper difficult to read, so we reduced the number of tables commenting explicitly their main peculiarities in the text. In addition, some results interpretations and conclusions have been cautiously reviewed and rewritten.

• The use of a dynamical thinning in relationship with the outer loop technique needs to be clarified (see my comment below). In the end, are there more or less radar data ingested with this technique?

We are very sorry for the misunderstanding created with the words "dynamical thinning" in relationship with the outer loop techinque: we didn't mean that thinning and outer loop strategy are the same thing, not at all! Because thinning is used to reduce the number of observations assimilated in NWP models, vice versa the outer loop strategy allowed the ingestion of more observations progressively.

**Specific comments**

• Section 2.1: It should be mentioned in the text that Figure 2 was produced with DEWETRA. Otherwise, the reader who overlooks Figure 2's caption does not understand why DEWETRA is introduced here.

We agree with the reviewer. The sentence has been modified as follows: " Figure 2, <mark>produced using DEWETRA operational platform</mark>, shows the interpolated map ....."

• Section 3.1: The authors explain that 'volume reflectivity radar data, for each elevation, are projected onto the Cartesian plane in order to find the closest radar bin for each Cartesian grid

point and then they are interpolated by the 3D-Var code of WRF'. This is still unclear to me. Does it mean that there is a radar observation assimilated at every model grid point (that of 'the closest radar bin')? What kind of interpolation is done by the 3D-Var code? In other words, the interesting (and missing) piece of information here is the *spatial resolution* of the observations.

We recognize that as it is written the phrase you mentioned could create a misunderstanding. Following we try to better explain the procedure performed on radar data and to clarify the sentence.

The first step in the radar data processing for assimilation involves transformation of radar data to geographical coordinates. Radar data are originally given in a polar geometry in which data points are represented in range, azimuth and elevation. Before assimilation, weather radar data volumes need to be transformed into geographical (latitude, longitude, altitude) Cartesian coordinates.

So, in this study radar observations (specifically reflectivities) have been converted from the polar coordinates to lat/lon and elevation, that is the observation information contained into the "ob.radar" file.

As stated by WRFHELP (the email assistance service that provides user support) WRFDA does not have any QC or pre-processing/smoothing/interpolation built in; "ob.radar" should contain observation information in terms of lat/lon and elevation. There should be no interpolation except to convert from the radar's native polar coordinates to lat/lon and elevation.

Anyway the sentence aforementioned has been modified as follows: <mark>"volume reflectivity radar data are converted from their native polar coordinates (range, azimuth and elevation) into geographical Cartesian ones (latitude, longitude and elevation)".</mark>

| Features                               | Units | MM                   | P55C                                        | SPC                                                   |
|----------------------------------------|-------|----------------------|---------------------------------------------|-------------------------------------------------------|
|                                        |       | radar                | radar                                       | radar                                                 |
| Owner                                  |       | CF Abruzzo
Region | ISAC-CNR of
Rome                         | <mark>Arpae</mark> Emilia Romagna                     |
| Location                               |       | Monte Midia          | Rome                                        | San Pietro Capofiume                                  |
| Latitude                               | [deg] | 42.057               | 41.840                                      | 44.6547                                               |
| Longitude                              | [deg] | 13.177               | 12.647                                      | 11.6236                                               |
| Height (a.s.l.)                        | [m]   | 1760                 | <mark>131</mark>                            | 31                                                    |
| Doppler                                |       | YES                  | YES                                         | YES                                                   |
| Dual Polarization               |       | NO                   | YES                                         | YES                                                   |
| Range Resolution                       | [m]   | 500                  | 75                                          | 250                                                   |
| Half Power Beam Width                  | [deg] | <mark>1.6</mark>     | 1                                           | <mark>0.9</mark>                                      |
| Temporal Resolution             | [min] | 15                   | 5                                           | 15                                                    |
| Elevations angles
used in PPI scans | [deg] | 0, 1, 2, 3           | 0.6, 1.6, 2.6, 4.4,
6.2, 8.3, 11.0, 14.6 | 0.53, 1.4, 2.3, 3.2, 4 .1,
<mark>5.0</mark> |
| Maximum Range                          | [km]  | 120 or 240           | 120                                         | 125                                                   |

 Table 1, on technical characteristics of the three radars, has been modified as follows:

Moreover, the missing piece of information about spatial resolution of the observations has been added to the manuscript as follows: "each radar has a half power beam width of 1.6, 1 and 0.9 degree respectively for Monte Midia (MM), Polar55C (P55C) and San Pietro Capofiume (SPC) and a range resolution of 500, 75 and 250 metres."

Section 3.1: The authors write: 'Moreover, no observation thinning is performed because this procedure is not yet developed into the 3D-Var system for radar data. Nevertheless, a dynamical thinning has been devised that selects, for every assimilation cycle, the most influential partition of a particular measurement, from information based on the previous cycle: this is the multiple outer loops technique explained later in Section 4.' I have a different understanding of the outer loop technique. I understand that it is meant to update linearised operators (such as the observation operator) during the minimization process. As a consequence, more observations are assimilated with each iteration and the quality of the analysis is improved.

I do not see the relationship between the outer loop technique and thinning. The purpose of the latter is to counterbalance the use of an overly simplistic (ie, diagonal) observation error covariance matrix or to reduce the computational cost of the assimilation. Thinning actually results in reducing the amount of observations.

So what is the 'dynamical thinning [that] has been devised that selects, for every assimilation cycle, the most influential partition of a particular measurement, from information based on the previous cycle'? In the cited literature, Rizvi et al. (2008) pertains to the outer loop technique (in passing, it may be more appropriate to cite peer-reviewed articles such as Hsiao et al. 2012), and Liu and Rabier (2002) pertains to thinning, but no citation refers to both thinning and outer loop.

As already anticipated in the general comments we are very sorry for creating the misunderstanding between thinning and outer loops, we exactly know that they are two different techniques with two different and opposite scopes. On one hand, data thinning techniques (like superobing) have both the goal to reduce the amount of data (extracting the essential information content and decreasing the computational cost of the assimilation) and spatial error correlations between adjacent observations. On the other hand, the outer loops strategy is used to solve the non-linear problems involved in the calculation of the observation operator: this technique results in an improvment of the quality of the analysis because the number of the observations assimilated increased for each subsequent outer loop, since the observations that were rejected in the previous outer loop can enter in the succeeding outer loop.

The manuscript has been revised in the light of these last considerations.

Lines 191-194 of the old version of the manuscript have been modified as follows: "Moreover, no observation thinning is performed because this procedure is not yet developed into the 3D-Var system for radar data. Instead, an iterative approach has been applied to extract more information from radar data during the assimilation procedure: this is the multiple outer loops technique explained later in Section 4".

We agree with the reviewer that Hsiao et al 2012 is more appropriate to cite for the outer loops technique, so we replaced Rizvi et al. 2008 with Hsiao et al. 2012 and lines 280-284 of the old version of the manuscript have been rewritten as follows: "Finally, an experiment to assess the role of the outer loops is performed (CONMMPOLSPC3OL): to include non-linearities into the observation operator and to evaluate the impact of reflectivity data entering for each cycle, the multiple outer loops strategy is applied (Hsiao et al. 2012). According to this approach, the non-linear problem is solved iteratively as a progression of linear problems: the assimilation system is able to ingest more observations by running more than one analysis outer loop, allowing observations rejected in the previous loop to be enter into the subsequent one. Since radar data are non linearly related to the analysis control variables, the outer loops method is particularly helpful to extract more information from such data."

In the "References" too, Rizvi et al. 2008 has been removed, whereas Hsiao et al. 2012 has been added.

Concerning Liu and Rabier 2002 (line 387 of the old version of the manuscript), the citation in the paper wanted to be pertain to thinning, not to both thinning and outer loop.

The sentence in question is almost a verbatim excerpt from Cardinali (2013, 2014). Does it mean that the authors used a dynamical thinning based on the influence matrix, which is the topic dealt with by Cardinali (2013, 2014)? In that case, they should add a reference to the technique they used or give more details about how it works. If this is related to the outer loop technique, the authors should formulate this relationship more explicitly.

The aforementioned sentence is not related to the influence matrix but to the outer loops approach. The sentence has been formulated more explicitly as follows: "Instead, an iterative approach has been applied to extract more information from radar data during the assimilation procedure: this is the multiple outer loops technique explained later in Section 4."

• Section 5, comments on Table 4: Table 4 contains a lot of figures and the conclusions which are drawn from it are that the values are 'good' for ACC and FAR (which is expected when the considered events are rare) and that the experiments overestimate light precipitation. Is Table 4 really needed?

We agree with the reviewer that Table 4 is not strictly necessary so it has been removed. Conclusions which are drawn from it have been summarized in the text as follows (lines 310-316 of the new version of the manuscript): "At an objective comparison of the statistical indices (not shown here) with their relative upper and lower confidence limits for the 12 hours accumulated precipitation and for two thresholds (1 mm and 40 mm for light and heavy rain regimes respectively), we obtained likely good values for ACC and FAR for all the experiments and for heavy rain regimes, strengthened by a small uncertainty interval. On the other hand, for the lower threshold the values of FBIAS for all simulations, considering also the confidence intervals, are greater than one. One possible interpretation of the impact of the lower threshold is that with 95% confidence all the experiments are overestimating the frequency of precipitation around 1 mm/12h."

• Section 5: What message do the authors want to convey with the following sentence: 'MET indices in Table 5 suggest that CTL and CON\_HR\_12KM have the widest spread between the CIs limits for higher thresholds'?

With the aforementioned sentence the authors want to say that CTL and CON\_HR\_12KM are the experiments with the largest difference between the confidence intervals bounds for heavy rain regimes of FBIAS. This result suggests that the remaining simulations, with smallest difference in CIs limits and with both bounds lower than 1, surely understimate the frequency of heavy precipitating events. Conversevely, we don't assert the same for CTL and CON\_HR\_12KM.

We modified the sentence as follows (lines 330-335 of the new version of the manuscript): "MET indices (not shown here) suggest that CTL and CON\_HR\_12KM have the largest difference between the CIs bounds for higher thresholds of FBIAS: this result suggests that the remaining simulations, with smallest difference in CIs limits and with both bounds lower than 1, surely underestimate the frequency of heavy precipitating events."

• Section 5, I 327: I do not understand how the conclusion that 'CONMMPOLSPC3OL\_HR\_12KM is the simulation with the best response' is reached. The score values for all experiments are quite close

to each other and within the uncertainty intervals, and CONMMPOLSPC3OL\_HR\_12KM even scores lower than CTL for ACC(1 mm), FBIAS and ETS(1 mm).

We agree with the reviewer that some results interpretations have to be give with caution as in this case where the score values for all simulations are quite close to each other. We modified the sentence as follows:"Another aspect to point out is that some indices for all simulations are quite close to each other and within the CIs, so it is not possible to discern which is the best experiment over all"

 Section 5, Il 338-339: I do not understand that 'the frequency of rainfall overestimation for higher thresholds has been reduced by radar reflectivity assimilation performed only on D01'. For higher thresholds, FBIAS is systematically below 1, which means that the experiments underestimate the frequencies of large rainfall accumulations. The underestimations are even worse when radar reflectivity data are assimilated in D01 only: all FBIAS score values lie below .31 when radar reflectivity is assimilated, vs .47 and .49 for CTL and CON\_HR\_12KM, respectively.

We agree with the reviewer, the sentence has been modified as follows: "the frequency of rainfall underestimation for higher thresholds found in the mother domain when radar reflectivity data are assimilated in D01 only has been reduced by switching to a higher resolution domain, moreover, the overestimation of the frequency for lower thresholds has been corrected because the FBIAS, previously systematically above 1, is found approximately 1 (indices not shown)."

• Section 5, Il 342-344: 'The assimilation, operated on both 12 km and 3 km, gives better results than the ones on column 1, but a worse response than the others on column 2 is given for higher thresholds.' Could the authors please back this up? It is far from straightforward to see it.

The aforementioned sentence has been clarified as follows: "Data assimilation, operated on both 12 km and 3 km, shows similar performances to the experiments where assimilation is performed only on D01 (table 4), but a worse response for higher thresholds (tables 3 and 4) than the ones where assimilation is carried out on D02."

• Section 5, Il 378-380: How can shielded radar data lead to underestimating precipitation forecasts? I understood that they had been filtered out (see Il 184-185 'all the data that are affected by partial beam blocking and clutter have been filtered out').

Radar shielding depends on the location of the radar with respect to surrounding hills and mountains. Consider if a beam was filtered out along some azimuth angles, only beams at higher elevation angles are assimilated and at longer range only the ice regions are samples, leading to a partial distribution of the 3D reflectivity.

The aforementioned sentence has been rewritten as follows: "With regard to the assimilation of reflectivity radar data, it should be noted that P55C radar observations of the event considered is shielded at the lowest elevation angles by the Apennines range and provides a limited contribution to reflectivity data that are assimilated."

**Impact of Multiple Radar reflectivity data assimilation on the numerical simulation of a Flash Flood Event during the HyMeX campaign**

Ida Maiello1,2, Sabrina Gentile3,1, Rossella Ferretti2, Luca Baldini4, Nicoletta Roberto4, Errico Picciotti5,2, Pier Paolo Alberoni6, Frank Silvio Marzano1,2

[revised manuscript text omitted]

---

## Author Response (AR4)

**Review of 'Impact of Multiple Radar reflectivity data assimilation on the numerical simulation of a Flash Flood Event during the HyMeX campaign' by Maiello et al.**

**General comments**

The manuscript has been substantially improved and I recommend it to be published once the remaining minor comments here below are addressed.

*We think too that the manuscript has been really improved compared to the beginning of the review thanks to the fruitful referees' and Editor's comments that have addressed shortcomings and weak points. Following remaining minor comments have been addressed and also typos, co-authors and their affiliations, terminology, data in tables and variables in equations have been carefully checked. All changes in the manuscript have been marked-up in yellow.*

**Specific comments**

- Is 'a non-Gaussian error probability density function' (ll 74-75) the reason why the 'simulation' (l 73) of radar reflectivity is challenging? The authors maybe meant 'assimilation' instead of 'simulation'.
  *Yes, first of all we meant "assimilation" instead of "simulation" in the aforementioned sentence. The formulation of the observation operator for the radar reflectivity is not as straightforward, because it depends on the assumption of drop size distribution in a microphysical parameterization scheme and the classification of hydrometeors. The challenge stays in the non-linearity and non-Gaussian error pdf.*
  *The fact is, with real data you don't know what the probability distribution of the errors is, and you don't even know that it has any particular mathematical form that is consistent from one experiment to another.*

- l 194: Is 'altitude' meant here instead of 'elevation'? I suppose the WRF3D-Var system does not know about the radar elevation angles, position, etc.
  *That's right! We meant "altitude" here instead of "elevation". Any specific radar characteristic is not a priori known by WRF3D-Var system.*

- ll 288-292: The answers of the authors to my previous review partly clarified how the radar data are processed in the assimilation system.
  I understand that all radar observations enter the WRF-3D-Var system. Some of them are rejected, I suppose based on observation-minus-guess departures. The outer loop technique allows to increase the amount of assimilated data at each iteration.
  Could the authors give an example or order of magnitude of: i) the amount of radar data that enter the 3D-Var system, ii) the fraction of radar data that are rejected, say, at the first

iteration and at the last one, respectively? That would show the efficiency of the outer loop technique in assimilating more data and give keys to understand the differences between experiments using this technique or not.

*The total amount of radar data (considering the experiment where all the three radars are assimilated) that initially enter the 3D-Var system is 518400.*

*As an example, the fraction of radar observations assimilated into the 3 km domain at the first outer iteration is 32986, at the second outer iteration is 33001 and at the last one is 33027. As you can understand, more than 480000 radar observations are rejected at the first external iteration but you can see, as the number of outer loops increased, a small number of data enter in the subsequent ones. Note that the maximum permissible value for the setting of outer loops in WRF 3D-Var is 10; for this study we used 3 outer loops as suggested by literature and to demonstrate how some lost data have been recovered at each outer loop.*

*The aforementioned information are also inserted in section 4 of the manuscript as follows:*
*"For example, over a total amount of 518400 radar data (considering all the three radars), the fraction of radar observations assimilated into the 3 km domain at the first outer iteration is 32986, at the second outer iteration is 33001 and at the last one is 33027."*

- Seven tables of verification statistics (out of nine in the previous version) are still present in this new version. I leave it to the editor to decide whether it is acceptable or should either be reduced or converted into figures for improved legibility.

  *We are very sorry both with the referee and the Editor if the number of tables of verification statistics is still too high and weaken the legibility of the manuscript. Therefore we decided to merge Table 3 with Table 4 (that becomes new Table 3 in the new version of the manuscript), Table 5 with Tables 6,7,8 (new Table 4) and old Table 9 becomes new Table 5. Therefore the number of tables of verification statistics has passed from seven to three as follows:*

[revised manuscript text omitted]

**0.83**
(0.86) | (0.97)
**0.98**
(0.99) | (0.82)
**0.98**
(1.18) | (0.08)
**0.15**
(0.24) | (0.19)
**0.30**
(0.41) | (0.04)
**0.11**
(0.18) | (0.19)
**0.25**
(0.31) | (0)
**0.002**
(0.003) |